# Revitalizing Channel-dimension Fourier Transform for Image Enhancement

## Abstract

Fourier transform and its variants have made impressive progress in image enhancement, benefiting from their capability for global representation. However, previous works mainly operate in the spatial dimension, potentially under-exploring the discriminative features inherent in the channel dimension. In this study, we introduce the channel-dimension Fourier transform for image enhancement, where the transformation is applied to channel-wise representation to enhance its discrimination ability of global representation. Specifically, we offer three alternative implementations of the channel transform, performing operations in *1) the global vector with its higher order moment*, *2) the global vector divided by groups in channel dimension*, and *3) the Fourier features derived from spatial-based Fourier transform*. The above fundamental designs, serving as generic operators, can be seamlessly integrated with existing enhancement network architectures. Through extensive experiments across multiple image enhancement tasks, such as low-light image enhancement, exposure correction, SDR2HDR translation, and underwater image enhancement, our proposed designs consistently demonstrate performance gains. The code will be made publicly available.

## 1 Introduction

Image enhancement aims to recover a clear image from its degraded counterpart captured under unfavorable light conditions or severe environments. Representative image enhancement tasks include low-light image enhancement, exposure correction, SDR2HDR translation, and underwater image enhancement, among others. The performance of image enhancement affects not only visual quality but also the applications of computer vision techniques.

Deep learning-based methods have witnessed remarkable advancements in image enhancement and shown powerful capability in modeling lightness and contrast adjustment procedures. A line of works customizes degradation prior-aware paradigms to explicitly learn the lightness component, such as curve-adjustment (Guo et al., 2020) and Retinex theory-based methods (Wei et al., 2018). These studies typically divide the learning process into global and local components and may not fully capture the dependencies within the feature space. In addition, another line of research concentrates on roughly designing complex networks to implicitly learn lightness and contrast enhancement procedures (Xu et al., 2022). However, these approaches have not deeply explored the underlying mechanism of image enhancement or introduced dedicated operations for handling global components, thus constraining their ability to effectively learn lightness and contrast adjustment.

Fourier transform has demonstrated its effectiveness in global information modeling (Chi et al., 2020). As a supplement to the aforementioned works, some endeavors have integrated Fourier transform-based operations into image enhancement architectures to adjust global components (Li et al., 2023; Huang et al., 2022). By employing Fourier transform on the spatial dimension, which yields global statistical information for each channel, this operation improves the distinguishability of various global representations and streamlines their learning process. Despite its effectiveness, we argue that the discriminability of global representation can also be addressed by modeling the channel distribution since the Gram matrix that connects the channel dimension information has shown its advantages in modeling global style information. Learning in the channel-dimension space using Fourier transform can enhance the discriminability of global representations and thus

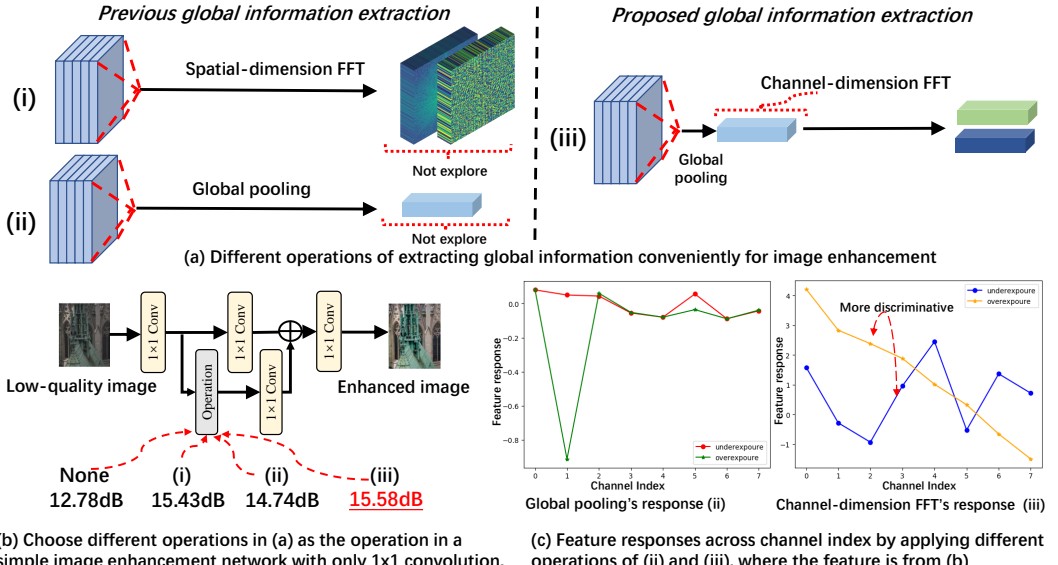

Figure 1: Motivation. (a) presents the different global representation formats (spatial-dimension FFT, global pooling, our channel-dimension FFT) for image enhancement. (b) and (c) are conducted in exposure correction on the SICE dataset. As shown in (b), employing a simple network coupled with our proposed channel-dimension Fourier transform outperforms the compared global formats, making our method a perfect fit for image enhancement tasks. (c) provides evidence that the channel feature response of our channel-dimension Fourier transform is more discriminative across different lighting conditions compared to the response curve of the compared global formats.

contribute to performance improvement. We present an example to show its ability of modeling global information in Fig. 1.

In this work, we propose a novel Channel-dimension Fourier transform learning (CFTL) mechanism for image enhancement. The primary objective of CFTL is to capture global discriminative representations by modeling channel-dimension discrepancy in an efficient global vector-based space. We implement the simple yet effective CFTL with three straightforward steps: (1) Apply the Fourier transform to the global averaged vector in the channel dimension; (2) Derive the amplitude and phase components of the transformed features and perform the channel-wise modulation on them; (3) Convert the obtained feature back to its original space through an inverse Fourier transform in the channel dimension and then add it to the spatial features. As depicted in Fig. 1 (b), by employing Fourier transform over channel dimension, the simple network suggests the strongest ability for fitting lightness adjustment, and the discriminability among different brightness representations is improved, as shown in Fig. 1 (c). Therefore, applying the Fourier transform on the channel dimension brings an effective space for adjusting the global representation, which can contribute to image enhancement tasks.

Following the above observations and analysis, we provide several implementation formats of the channel-dimension Fourier transform in different spaces, as shown in Fig. 3: (1) Performing operations in the global vector space with its different moment orders; (2) Performing operations on the global vector divided by groups in channel dimension; (3) Performing operations on the Fourier features converted by spatial-based Fourier transform. Across various image enhancement tasks, our extensive experiments consistently demonstrate performance improvements achieved by incorporating CFTL into existing image enhancement network architectures. Furthermore, CFTL also contributes to the development of a lightweight backbone, achieving an elegant balance between effectiveness and efficiency.

Our contributions are summarized as follows: **(1)** We provide a new mechanism for image enhancement tasks through revitalizing channel-dimension Fourier transform learning. This mechanism enhances the discriminability of global representation through channel statistics modeling, which further acts as a representative space to efficiently adjust global information. **(2)** We provide dif-

ferent applicable formats of the Channel-dimension Fourier transform learning (CFTL) mechanism, and conduct extensive experiments in various image enhancement tasks, demonstrating its potential in wide-range applications. **(3)** Our proposed CFTL is compatible with existing image enhancement network architectures, leading to performance improvement with negligible computation costs.

## 2 RELATED WORK

**Image Enhancement.** Image enhancement aims to improve the quality of low-visibility images by adjusting the global lightness and contrast components, *i.e.,* illumination, color, and dynamic range. Different image enhancement tasks play different roles in various scenarios. In low-light image enhancement, algorithms are tailored to enhance the visibility of images acquired in low-light conditions (Chen et al., 2018; Zhang et al., 2021). For exposure correction, methods concentrate on correcting images captured under both underexposure and overexposure scenes to normal exposure (Yang et al., 2020; Afifi et al., 2021). For SDR2HDR translation, this task aims to convert images from a low-dynamic range to a high-dynamic range (Chen et al., 2021a; He et al., 2020). For underwater image enhancement, the contrast and color need to be adjusted (Li et al., 2019b). Since recent approaches in image enhancement tasks leverage deep neural networks, adjusting global information (i.e., lightness, contrast) in an efficient space would effectively improve performance.

**Fourier transform.** Fourier transform is a popular technique for frequency domain analysis. This transformation shifts the signal to a domain with global statistical properties and is consequently utilized for various computer vision tasks. Fourier transform is a classic application extensively used for domain generalization and adaptation because of its effective modeling of global information. For instance, Xu *et.al* (Xu et al., 2021a) implement a Fourier-based data augmentation strategy to generate samples with diverse styles for domain generation. Lee *et.al* (Lee et al., 2023) propose to improve the normalization for domain generalization by recomposing its different components in the Fourier domain. In another application, the Fourier transform mechanism is utilized to design effective backbones, leveraging its ability to capture global information. For example, FFC (Chi et al., 2020) is introduced to process partial features in the Fourier domain, enabling models to possess a non-local receptive field. Besides, GFNet (Rao et al., 2021) utilizes FFT/IFFT to extract Fourier domain features, serving as global filters for effective attention modeling. All the above works demonstrate the effectiveness of Fourier domain features in capturing global spatial statistics.

More recently, Fourier transform has been introduced to low-level vision tasks (Fuoli et al., 2021; Mao et al., 2023). As an early attempt, Fuoli *et al* (Fuoli et al., 2021) propose a Fourier transform-based loss to optimize the global high-frequency information for efficient image super-resolution. DeepRFT (Mao et al., 2023) is proposed for image deblurring, which captures both low-frequency and high-frequency properties of various blurs with a global-receptive field, and a similar design is also employed for image inpainting (Suvorov et al., 2022). FECNet (Huang et al., 2022) suggests that the amplitude of the Fourier feature decouples the global lightness components and thus is effective for image enhancement. Yu *et al* (Yu et al., 2022) also observes a similar phenomenon in image dehazing, in which the amplitude reflects the global haze-related information.

## 3 METHOD

In this section, we first revisit the traditional 2D Fourier transform, followed by the introduction of our channel-dimension Fourier transform. Subsequently, we investigate the details of the proposed CFTL. Finally, we illustrate the implementation of the CFTL variants.

### 3.1 PRELIMINARY OF FOURIER TRANSFORM

As recognized, Fourier transform is widely used to analyze the frequency representation of images. Typically, this operation is independently conducted over the spatial dimension of each individual channel. Given an image $x \in \mathbb{R}^{H \times W \times C}$, the Fourier transform $\mathcal{F}(\cdot)$ converts it to Fourier space, obtaining the complex component $\mathcal{F}(x)$, which is expressed as:

$$\mathcal{F}(x)(u,v) = \frac{1}{\sqrt{HW}} \sum_{h=0}^{H-1} \sum_{w=0}^{W-1} x(h,w) e^{-j2\pi(\frac{h}{H}u + \frac{w}{W}v)}, \tag{1}$$

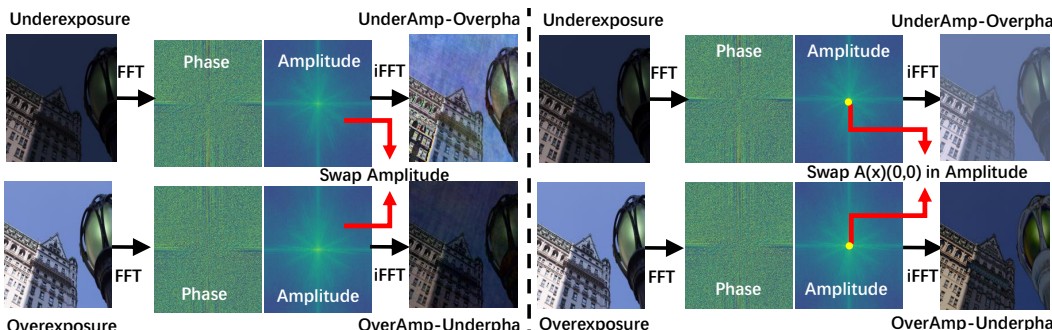

Figure 2: The illustration of swapping amplitude component in Fourier domain. Both swapping the amplitude swapping $\mathcal{A}(x)(0,0)$ lead to lightness swapping, while the latter produces fewer artifacts.

Both the Fourier transform and its inverse procedure $\mathcal{F}^{-1}(\cdot)$ can be efficiently implemented by FFT/IFFT algorithms (Frigo & Johnson, 1998). The amplitude component $\mathcal{A}(x)(u,v)$ and phase component $\mathcal{P}(x)(u,v)$ are expressed as:

$$\mathcal{A}(x)(u,v) = \sqrt{R^2(x)(u,v) + I^2(x)(u,v)},$$
$$\mathcal{P}(x)(u,v) = \arctan[\frac{I(x)(u,v)}{R(x)(u,v)}], \tag{2}$$

where $R(x)(u,v)$ and $I(x)(u,v)$ represent the real and imaginary part, respectively. When referring to the amplitude component $\mathcal{A}(x)(u,v)$, it quantifies the magnitude of the frequency index $(u,v)$ in an image, serving as a statistical indicator of frequency.

Targeting image enhancement, previous works have demonstrated that global information such as lightness is mainly preserved in the amplitude component (Li et al., 2023). However, we argue that the primary characteristic of global information remains conserved within $\mathcal{A}(x)(0,0)$, as shown in Fig. 2. The example represents using the low-frequency part in the amplitude has already led to effective global information swapping, while using the whole amplitude may bring artifacts in the swapped result. With regard to Eq. 2, the globally averaged vector across the 2D dimension is equal to $\mathcal{A}(x)(0,0)$, and we utilize it as the subject of manipulation.

In addition, different channels exhibit different properties of spectral information, which also determine the global information of an image when conjunct different channels.

A comparable inference can be drawn from style transfer studies, wherein the Gram matrix signifies global style information (Li et al., 2017). This inspires us to employ the Fourier transform on the channel dimension to enrich the representation of global information, as detailed below.

## 3.2 CHANNEL-DIMENSION FOURIER TRANSFORM

We introduce the channel-dimension Fourier transform by individually applying Fourier transform along the channel dimension for each spatial position. For each position $(h \in \mathbb{R}^{H-1}, w \in \mathbb{R}^{W-1})$ within the feature tensor $x \in \mathbb{R}^{H \times W \times C}$, denoted as $x(h,w,0:C-1)$ and abbreviated as $y(0:C-1)$, Fourier transform $\mathcal{F}(\cdot)$ converts it to Fourier space as the complex component $\mathcal{F}(y)$, which is expressed as:

$$\mathcal{F}(y(0:C-1))(z) = \frac{1}{C} \sum_{c=0}^{C-1} y(c) e^{-j2\pi \frac{c}{C} z}. \tag{3}$$

Here, the amplitude component $\mathcal{A}(y(0:C-1))(z)$ and phase component $\mathcal{P}(y(0:C-1))(z)$ of $\mathcal{F}(y(0:C-1))(z)$ are expressed as:

$$\mathcal{A}(y(0:C-1))(z) = \sqrt{R^2(y(0:C-1))(z) + I^2(y(0:C-1))(z)},$$
$$\mathcal{P}(y(0:C-1))(z) = \arctan[\frac{I(y(0:C-1))(z)}{R(y(0:C-1))(z)}]. \tag{4}$$

These operations can also be applied for the global vector $x_g \in \mathbb{R}^{1 \times 1 \times \mathrm{C}}$ derived by the pooling operation (see Eq. 5). In this way, $\mathcal{A}(y)(z)$ and $\mathcal{P}(y)(z)$ signify the magnitude and directional changes in the magnitude of various channel frequencies, respectively. Both of these metrics encapsulate global statistics related to channel information.

We provide visualization to suggest the properties of this operation in Fig. 16. It is evident that distinct representations of lightness become more discernible following the channel-based Fourier transform, both in terms of $\mathcal{A}(y)(z)$ and $\mathcal{P}(y)(z)$. This indicates that this operation improves the distinguishability of global information components, and adjusts the channel statistics would significantly influence its properties. Therefore, the transformed feature can act as a representative space for global information adaptation.

### 3.3 CHANNEL-BASED FOURIER TRANSFORM LEARNING

Based on the above analysis, we introduce the CFTL implementation, as shown in Fig. 3, which conducts operations on channel-based Fourier transformed features.

**Operation Description**. The core construction of the CFTL involves a three-step sequential process: applying the Fourier transform to the channel dimension to obtain channel-wise Fourier domain features, performing a channel-wise transformation on both its amplitude and phase components, and then reverting back to the spatial domain.

Given the feature $x \in \mathbb{R}^{\mathrm{H} \times \mathrm{W} \times \mathrm{C}}$, the initial step involves transforming it into a global vector $x_g \in \mathbb{R}^{1 \times 1 \times \mathrm{C}}$ using global average pooling as:

$$x_g = \frac{1}{\mathrm{HW}} \sum_{h=0}^{\mathrm{H}-1} \sum_{w=0}^{\mathrm{W}-1} x(h, w). \tag{5}$$

Here, $x_g$ equals $\mathcal{A}(x)(0,0)$ as described above, effectively encapsulating global information. Then, $x_g$ is transformed into channel-dimension Fourier domain using Eq. 3, denoted as $\mathcal{F}(x_g)(z)$.

Secondly, we utilize Eq. 3 to transform the feature $\mathcal{F}(x_g)(z)$ into its amplitude component $\mathcal{A}(x_g)(z)$ and phase component $\mathcal{P}(x_g)(z)$. Instead of acting upon $\mathcal{F}(x_g)(z)$, we advocate performing operations on $\mathcal{A}(x_g)(z)$ and $\mathcal{P}(x_g)(z)$ due to their explicit information meaning. Conversely, $\mathcal{F}(x_g)(z)$ lacks the requisite discriminative properties, as detailed in the Appendix. We introduce attention-based operations on $\mathcal{A}(x_g)(z)$ and $\mathcal{P}(x_g)(z)$

$$\begin{aligned}
\mathcal{A}(x_g)(z)' &= \mathtt{Seq1}(x_g) \odot \mathcal{A}(x_g)(z), \\
\mathcal{P}(x_g)(z)' &= \mathtt{Seq2}(x_g) \odot \mathcal{P}(x_g)(z),
\end{aligned} \tag{6}$$

where $\mathtt{Seq1}(\cdot)$ and $\mathtt{Seq2}(\cdot)$ denote sequences of $1 \times 1$ convolutions followed by LeakyReLU activation, as illustrated in Fig. 3. The symbol $\odot$ signifies element-wise multiplication for attentive adjustment. Hence, Eq. 6 signifies the process of modifying the global information encapsulated within the channel statistics $\mathcal{A}(x_g)(z)$ and $\mathcal{P}(x_g)(z)$.

Finally, we convert the processed channel-dimension Fourier domain feature $\mathcal{A}(x_g)(z)'$ and $\mathcal{P}(x_g)(z)'$ to their original space by employing the inverted channel-based Fourier transform

$$x_g' = \mathcal{F}^{-1}(\mathcal{A}(x_g)(z)', \mathcal{P}(x_g)(z)'), \tag{7}$$

where $x_g'$ is the final processed feature of CFTL. Furthermore, we resize it by replication to the original resolution $\mathrm{H} \times \mathrm{W} \times \mathrm{C}$ to align with the size of $x$, as illustrated in Fig. 3.

**Integrating CFTL into backbone architectures.** Upon the processed feature $x_g'$, we integrate it with the processed original feature $x$, rendering CFTL compatible with existing backbone architectures. Given the resolution disparity between $x_g'$ and $x$, we initially expand $x_g'$ by repeating it by $\mathrm{H} \times \mathrm{W}$ times to match the size of $x$. Subsequently, we integrate $x_g'$ with the processed $x$. As depicted in Fig. 3, within the CNN-based backbones, $x$ undergoes a local information branch (*i.e.,* convolutional layers) for local information processing.

### 3.4 VARIANTS IMPLEMENTATION OF CFTL

Following the above rules, we offer three alternative implementation formats of the channel transform in different operational spaces on global vector $x_g$.

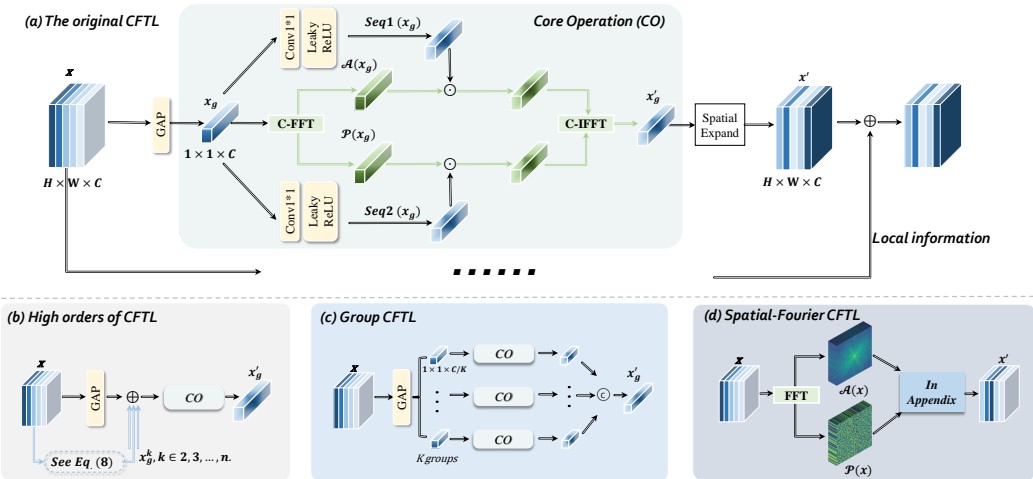

Figure 3: The illustration of the CFTL operation and its variants. (a) is the original CFTL, (b) is the High-order CFTL, (c) is the Group CFTL, and (d) is the Spatial-Fourier CFTL.

**Original CFTL.** We construct the original CFTL by treating $x_g$ as the global averaged vector of $x \in \mathbb{R}^{H \times W \times C}$, as shown in Fig. 3, which is plug-and-play for enhancement networks.

**High-order CFTL.** Regarding the global average vector $x_g$ in Eq. 5 as the first-order global information from $x$, we can introduce more orders to strengthen the representation ability of $x_g$, as illustrated in Fig. 3 (b). Specifically, for the $k$-th order of the global information, we denote it as $x_g^k$:

$$x_g^k = \sqrt[k]{\frac{1}{HW} \sum_{h \in [1,H]} \sum_{w \in [1,W]} (x(h,w) - x_g)^k}, \quad k \in [2,3,...,+\infty]. \tag{8}$$

This formula is derived from the color moment (Huang et al., 2010), which is another representation of global information. For example, with $k = 2$, $x_g^k$ signifies the standard deviation of $x$, we set $k$ to 2 by default in this format (as ablated in the Appendix). We further introduce $x_g^k$ to $x_g$ as its strengthened version:

$$x_g = x_g + x_g^2 + ...x_g^k, \quad k \in [2,3,...,+\infty]. \tag{9}$$

We process $x_g$ following similar operations in CFTL, considering it as the higher orders of CFTL.

**Group CFTL.** Since the original operations in CFTL are conducted on all channels of $x_g \in \mathbb{R}^{1 \times 1 \times C}$, we can further process different groups of channel statistics in $x_g$, which is similar to the group normalization (Wu & He, 2018). Specifically, we divide $x_g$ into $K$ groups along the channel dimension, and each group has $\frac{C}{K}$ channels, which is expressed as:

$$x_g = [x_g(1), ..., x_g(k)|k \in 1, ..., K], \tag{10}$$

where $x_g(k)$ is the $k$-th channel group of $x_g$. As shown in Fig. 3 (c), in the following operations of the Group CFTL, different $x_g(k)$ are processed by the operations in the CFTL respectively with different weights in Eq. 10. Finally, these processed features $x_g(k)'$ are concatenated along the channel dimension:

$$x_g' = \texttt{cat}[x_g(1)', ..., x_g(k)'], \tag{11}$$

where $\texttt{cat}[\cdot]$ denotes the concatenate operation. In this way, the parameters of the Group CFTL are less than the original CFTL as shown in quantitative results, while keeping competitive performance.

**Spatial-Fourier CFTL.** All the above variants are based on the global vector $x_g$, which is a special case of $\mathcal{A}(x)(u,v)$. Here, we propose to extend $x_g$ as the amplitude component $\mathcal{A}(x)(u,v)$ converted by the spatial Fourier transform, and we denote it as Spatial-Fourier CFTL. Therefore, $x_g \in \mathbb{R}^{H \times W \times C}$ has the same shape as $x$, which also exhibits global representation.

As shown in Fig. 3, to process $x_g$, we follow the rules of the CFTL to process it and obtain the result $x_g'$ (due to unstable training, we discard the few operations, which are detailed in the Appendix and

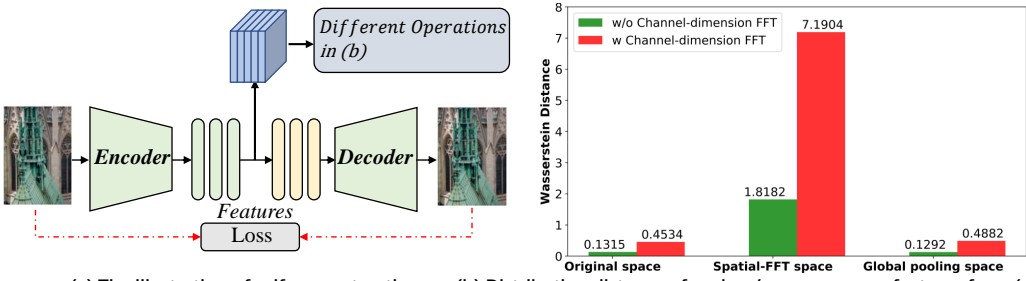

Figure 4: Toy experiment. (a) depicts the self-reconstruction that extracts the feature, which is processed by operations in Fig. 1. Evidenced by the feature similarity in (b), our proposed channel-dimension FFT enhances the discriminability between under- and over-exposure in different spaces.

shown in Fig. 10). Finally, $x'_g$ and $\mathcal{P}(x)(u, v)$ are converted back to the space of $x$ through the spatially-based inverse Fourier transform

$$x' = \mathcal{F}^{-1}(x'_g, \mathcal{P}(x)(u, v)), \tag{12}$$

where $x'$ represents the final result integrated into the processed $x$.

**CFTL-Net.** We present CFTL-Net, an efficient network architecture shown in Fig. 11, integrating High-order CFTL and Spatial-Fourier CFTL. This framework employs an encoder-decoder-based architecture, with additional details and discussions provided in the Appendix.

## 4 EXPERIMENT

To demonstrate the efficacy of our proposed CFTL paradigm, we conduct extensive experiments on several image enhancement tasks. More results can be found in the Appendix.

### 4.1 TOY EXPERIMENTS

To highlight the improved discriminability of global information using the channel-based Fourier transform, we conducted experiments, converting the image to the feature space(see Fig. 4). Specifically, we construct an encoder-decoder architecture for image reconstruction, trained on a dataset of 1000 samples sourced from the MIT-FiveK dataset (Bychkovsky et al., 2011). During testing, we used 100 underexposed and 100 overexposed samples from the SICE dataset as inputs, owing to their significant global information variations. We process the encoder-decoder features using various operations: spatial domain Fourier transform, global average pooling, and channel-based Fourier Transform on the global vector. Illustrations in the Appendix (see Sec. E) demonstrates that our proposed operation achieves the highest discriminability between underexposed and overexposed samples, as indicated by the maximum distribution distance.

### 4.2 EXPERIMENTAL SETTINGS

**Low-light image enhancement.** Following previous works (Hai et al., 2021; Zhao et al., 2021), we employ three widely used datasets for evaluation, including LOL dataset (Wei et al., 2018), Huawei dataset (Hai et al., 2021) and MIT-FiveK dataset (Bychkovsky et al., 2011). We employ two image enhancement networks, DRBN (Yang et al., 2020) and Restormer (Zamir et al., 2022) as baselines.

**Exposure correction.** Following (Huang et al., 2022), we adopt MSEC dataset (Afifi et al., 2021) and SICE dataset (Cai et al., 2018) for evaluations. Two architectures, *i.e.,* DRBN (Yang et al., 2020) and LCDPNet (Wang et al., 2022) are selected as baselines.

**SDR2HDR translation.** Following (Chen et al., 2021b), we choose the HDRTV dataset (Chen et al., 2021b) for evaluation. We employ CSRNet (He et al., 2020) as the baseline in the experiments.

**Underwater image enhancement.** Following prior works (Li et al., 2019b), we select UIEC^2-Net (Wang et al., 2021) as baseline and use UIEB (Li et al., 2019b) dataset for validation.

| Settings | #Param | FLOPs (G) | LOL | Huawei | FiveK |
|---|---|---|---|---|---|
| DRBN (Baseline) Yang et al. (2020) | 0.532M | 39.71 | 20.73/0.7986 | 19.93/0.6810 | 22.11/0.8684 |
| +Pooling attention | 0.533M (+0.01M) | 39.72 (+0.01) | 21.84/0.8176 | 20.13/0.6838 | 23.15/0.8702 |
| +Spaial Fourier | 0.533M (+0.01M) | 39.88 (+0.17) | 22.07/0.8355 | 20.28/0.6844 | 23.72/0.8735 |
| +Original CFTL | 0.534M (+0.02M) | 39.73 (+0.02) | **23.71/0.8492** | 20.82/0.6933 | **24.03**/0.8751 |
| +Group CFTL | 0.532M (+0M) | 39.71 (+0) | 22.98/0.8445 | 20.80/0.6952 | 23.93/**0.8768** |
| +High-order CFTL | 0.534M (+0.02M) | 39.73 (+0.02) | 23.05/0.8457 | 20.81/0.6930 | 23.95/0.8755 |
| +Spatial-Fourier CFTL | 0.536M (+0.04) | 40.27 (+0.56) | 22.31/0.8376 | **20.89/0.6954** | 23.96/0.8755 |
| Restormer (Baseline) (Zamir et al., 2022) | 26.10M | 563.96 | 20.49/0.7886 | 20.02/0.6663 | 23.13/0.8891 |
| +Pooling attention | 26.12M(+0.02M) | 569.41(+5.45) | 20.95/0.7952 | 20.34/0.6685 | 23.45/0.8915 |
| +Spatial Fourier | 26.12M(+0.02M) | 569.41(+5.45) | 21.01/0.8003 | 20.65/0.6713 | 23.66/0.8931 |
| +Original CFTL | 26.10M(+0M) | 563.97(+0.001) | 21.34/0.8020 | 20.70/0.6706 | 23.87/**0.8959** |
| +Group CFTL | 26.10M(+0.01M) | 563.97(+0.01) | **21.49**/0.8008 | 20.59/0.6715 | **23.91**/0.8942 |
| +High-order CFTL | 26.10(+0M) | 563.97(+0.01) | 21.27/**0.8061** | **20.76/0.6733** | 23.81/0.8944 |
| +Spatial-Fourier CFTL | 26.11(+0.01M) | 565.19(+1.23) | 21.37/0.8018 | 20.71/0.6713 | 23.82/0.8948 |
| CFTL-Net | 0.028M | 3.64 | 22.50/0.8139 | 20.91/06941 | 24.03/0.8904 |

Table 1: Quantitative results of low-light image enhancement in terms of PSNR/MS-SSIM.

| Settings | #Param | FLOPs (G) | MSEC | SICE |
|---|---|---|---|---|
| DRBN (Baseline) | 0.532M | 39.71 | 19.52/0.8309 | 17.88/0.6798 |
| +Pooling attention | 0.533M | 39.72 | 22.89/0.8604 | 20.75/0.7095 |
| +Spaial Fourier | 0.533M | 39.88 | 22.94/0.8642 | 20.94/0.7036 |
| +Original CFTL | 0.534M | 39.73 | **23.34/0.8683** | 21.32/**0.7250** |
| +Group CFTL | 0.532M | 39.69 | 23.33/0.8672 | 21.30/0.7177 |
| +High-order CFTL | 0.534M | 39.73 | 23.19/0.8667 | **21.64**/ 0.7243 |
| +Spatial-Fourier CFTL | 0.536M | 40.27 | 23.04/0.8645 | 21.33/0.7201 |
| LCDPNet (Baseline) | 0.961M | 9.40 | 22.30/0.8552 | 20.46/0.6843 |
| +Pooling attention | 0.962M | 9.41 | 22.41/0.8568 | 20.57/0.6835 |
| +Spaial Fourier | 0.962M | 9.43 | 22.47/0.8561 | 20.94/0.6946 |
| +Original CFTL | 0.962M | 9.41 | 22.68/0.8572 | 21.25/**0.7063** |
| +Group CFTL | 0.961M | 9.40 | 22.70/**0.8579** | 20.96/0.6952 |
| +High-order CFTL | 0.962M | 9.41 | **22.74**/0.8565 | **21.38**/0.7024 |
| +Spatial-Fourier CFTL | 0.967M | 9.48 | 22.52/0.8563 | 20.58/0.6865 |
| CFTL-Net | 0.028M | 3.64 | 22.88/0.8594 | 21.24/0.6999 |

Table 2: Results of exposure correction.

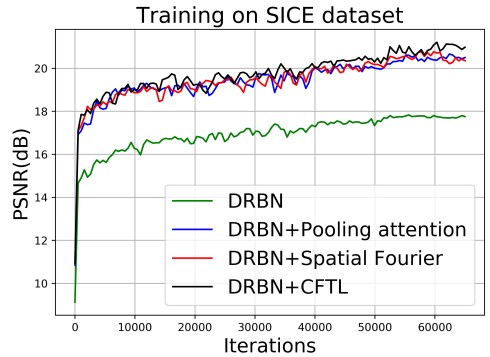

Figure 5: Training on exposure correction.

**Comparison operators.** We set two comparison operators corresponding to (i) and (ii) in Fig. 1, based on global pooling and spatial FFT. We refer to them as "Pooling Attention" and "Spatial Fourier", as illustrated in Sec. J in the Appendix, following the format of CFTL.

### 4.3 IMPLEMENTATION DETAILS

Given the three CFTL formats in Sec. 3.4, we integrate them individually into the baseline for experiments. For comparison, we conduct experiments with baseline networks and the integration of global average pooling and spatial Fourier transform. Furthermore, we include experiments with the CFTL-Net from Sec. 3.4. We train all baselines and their integrated formats using their original settings until they are converged. More implementation details are provided in the Appendix.

### 4.4 COMPARISON AND ANALYSIS

**Quantitative Evaluation.** We perform quantitative comparison on the image enhancement tasks in Table 1, Table 2, Table 3, and Table 4, where the best results are highlighted in bold. From the results, it can be observed that all formats of our proposed paradigm improve the performance of the baselines across the datasets in all tasks, validating the effectiveness of our proposed method. In contrast, applying the pooling attention operation or the Spatial Fourier transform cannot surpass our proposed CFTL in most datasets. Moreover, the proposed CFTL-Net also achieves effective performance with relatively fewer parameters. We also demonstrate that the proposed CFTL helps improve the training performance as shown in Fig. 5, where our method achieves higher PSNR during the training stage. Note that all the above evaluations show the effectiveness of applying our CFTL while introducing less computation cost.

**Qualitative Evaluation.** We present the visual results of exposure correction on the SICE dataset due to the limited space. As shown in Fig. 6, the integration of the CFTL leads to a more visually pleasing effect with less lightness and color shift problems compared with the original baseline. We provide more visual results in the Appendix.

| Settings | PSNR/SSIM | #Param/FLOPs(G) |
|---|---|---|
| CSRNet (Baseline) | 35.34/0.9625 | 0.035M/1.58 |
| +Pooling attention | 35.67/0.9635 | 0.045M/1.62 |
| +Spaial Fourier | 35.59/0.9647 | 0.048M/1.63 |
| +Original CFTL | 35.93/**0.9712** | 0.045M/1.62 |
| +Group CFTL | 35.82/0.9703 | 0.044M/1.62 |
| +High-order CFTL | 35.97/0.9694 | 0.045M/1.62 |
| +Spatial-Fourier CFTL | 35.74/0.9654 | 0.048M/1.62 |
| CFTL-Net | **37.37**/0.9683 | 0.028M/3.64 |

Table 3: Results of SDR2HDR translation.

| Settings | PSNR/SSIM | #Param/FLOPs(G) |
|---|---|---|
| UIEC^2-Net (Baseline) | 21.39/0.8957 | 0.53M/104.25 |
| +Pooling attention | 21.46/0 .8996 | 0.58M/113.93 |
| +Spaial Fourier | 21.63/0.9012 | 0.58M/113.93 |
| +Original CFTL | 21.81/0.9005 | 0.54M/104.27 |
| +Group CFTL | 22.20/**0.9042** | 0.54M/104.27 |
| +High-order CFTL | **22.33**/0.9023 | 0.54M/104.27 |
| +Spatial-Fourier CFTL | 22.26/0.9030 | 0.55M/106.43 |
| CFTL-Net | 22.27/0.9034 | 0.028M/3.64 |

Table 4: Results of underwater image enhancement.

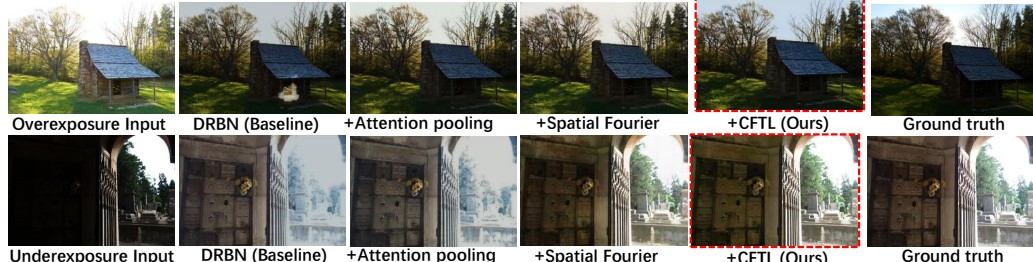

Figure 6: The visualization results on the SICE dataset for exposure correction.

| Configurations | SICE dataset | LOL dataset |
|---|---|---|
| Baseline (DRBN) | 17.88/0.6798 | 20.73/0.7986 |
| +CFTL | **21.32/ 0.7250** | **23.71/0.8492** |
| +CFTL w/o global pooling | 18.24/0.6853 | 19.91/0.8317 |
| +CFTL w/o channel-based ifft | 21.05/0.7228 | 22.66/0.8453 |
| +CFTL w/o processing amplitude | 21.21/0.7194 | 22.78/0.8438 |
| +CFTL w/o processing phase | 20.94/0.7201 | 23.29/0.8445 |

Table 5: Impact of CFTL configuration on SICE and LOL datasets in terms of PSNR/MS-SSIM.

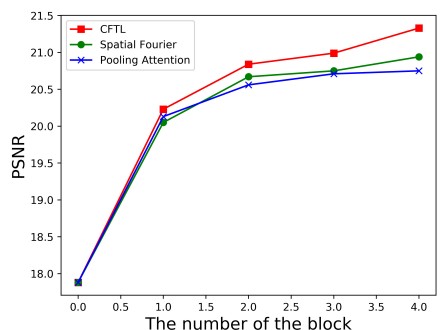

Figure 7: Impact of CFTL numbers on the SICE dataset.

## 4.5 ABLATION STUDIES

We conduct ablation studies on the exposure correction task and low-light image enhancement task using DRBN as the baseline. More ablation studies can be found in the Appendix.

**Investigate the design of CFTL.** To explore the design of CFTL, we perform experiments by setting the CFTL with different configurations. The quantitative results are shown in Table 5. As depicted, introducing the global pooling in the CFTL leads to significant performance improvement. Meanwhile, converting the feature to the original space with channel-dimension IFFT also works well. Note that conducting the operation on either the amplitude or phase components leads to sub-optimal results due to incomplete use of them. All results depict the reasonableness of our designs.

**Impact of the CFTL number.** We further investigate the impact of the CFTL numbers on the exposure correction task. The corresponding quantitative number $K$ comparison from 1 to 4 is reported in Fig. 7. As depicted, only incorporating one CFTL produces a significant performance improvement. By increasing the number of CFTL, the results are further improved apparently than other comparison operations, which can be attributed to its superior ability to adjust global information.

## 5 CONCLUSION

In this paper, we introduce a novel channel-based Fourier transform learning mechanism for image enhancement. The proposed CFTL enhances the discriminative capability of global information while acting as a representative space for global information adjustment. CFTL offers multiple implementation formats, easily integrated into existing image enhancement architectures with limited computational costs. Extensive experiments demonstrate the effectiveness and scalability of applying the CFTL and its variants across diverse image enhancement tasks.

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

# Appendix

In this appendix, we provide additional details and results.

In Sec. A, we present more implementation details.

In Sec. B, we extend the CFTL to other tasks, including image dehazing and pan-sharpening.

In Sec. C, we present more illustrations of the global information for image enhancement.

In Sec. D, we provide more illustrations and discussions about our work's motivation.

In Sec. E, we present more discussions about the toy experiment.

In Sec. F, we present more illustrations of the mechanism of our proposed method.

In Sec. G, we present the discussions about our work, including the reasons for its design formats, the reasons for its effectiveness, its limitations, and potential extension formats. Moreover, we supplement why the "channel-dimension discrepancy" helps improve image enhancement. We also provide more tasks related to image enhancement.

In Sec. H, we provide more discussions and results about the generalization ability of our method.

In Sec. I, we provide more discussions about different implementation formats.

In Sec. J, we present detailed information about the experimental settings.

In Sec. K, we provide more discussions about other network architectures and operations.

In Sec. L, we present more ablation studies to investigate the CFTL.

In Sec. M, we present more results of applying the CFTL in other backbones.

In Sec. N, we present more comparison results.

In Sec. O, we show more visualization results on multiple image enhancement tasks. We also supplement more visual results.

## A  More Implementation Details

**Pseudo code of the CFTL.** For implementing the CFTL, we provide the pseudo-code of the original CFTL and High-order CFTL in Fig. 8 as well as the Group CFTL and Spatial-Fourier CFTL in Fig. 9.

**Illustration of Spatial-Fourier CFTL.** We present the detailed illustration of CFTL in Fig. 10. As described in the main body and Fig. 9, we do not apply the channel-based iFFT on the processed spatial-based Fourier amplitude FAS in Fig. 9. Instead, we directly convert it back to the original space with the spatial-based Fourier phase. **The reason** is that applying the channel-based iFFT on FAS would lead to unstable training and result in the "NAN" phenomenon. The instability issue may stem from the unstable gradient in two sequential inverse Fourier transforms.

Since the spatial-based Fourier component is an effective global information representation, exploring its channel-dimension representation with a channel-based Fourier transform would enhance the representation. Meanwhile, the spatial-based Fourier component is a generalized representation of global pooling as described in Sec. 3.3. Therefore, it is reasonable to apply the channel-based Fourier transform to the spatial-based Fourier component. Finally, our Spatial-based CFFT achieves better performance than the previous Spatial-based Fourier transform, suggesting its effectiveness.

**Implementation details of CFTL-Net.** We implement the CFTL-Net in an encoder-decoder architecture consisting of four scales with a feature channel number of 8. The implementation structure is illustrated in Figure 11. CFTL-Net employs Spatial-Fourier CFTL and High-order CFTL to formulate the basic unit for processing features in one scale.

Specifically, the basic unit consists of two parts borrowed from the transformer. The former focuses on processing features in the spatial dimension, while the latter aims to process channel-dimension information. We apply Spatial-Fourier CFTL in the former part and the High-order CFTL in the latter part. In each part, besides the two operations proposed in this paper, we introduce the half-

```
def CFTL(F):                       def High-order CFTL(F):

 # F: input with shape [N, C,       # F: input with shape [N, C,
    H, W]                              H, W]
  FG = GlobalPooling(F)             FG = GlobalPooling(F)
  # Calculation in Eq.5             # Calculation in Eq.5
  #FG: [N, C, 1, 1]                 #FG: [N, C, 1, 1]
                                    F^k = Korder(F)
  FA, FP = CFFT(FG)                 # Calculation in Eq.8
  # Applying Channel-based          #F^k: [N, C, 1, 1]
     FFT in Eq.3 and Eq.4           Fsum = FG+F^k
  # FA and FP are the derived       # Integrate FG and F^k in
     amplitude and phase              Eq.9
  FA = Seq1(FG)*FA
  FP = Seq2(FG)*FP                  FA, FP = CFFT(Fsum)
  # Process amplitude and           # Applying Channel-based
     phase in Eq.6                     FFT in Eq.3 and Eq.4
  FI = iCFFT(FA,FP)                 FA = Seq1(FG)*FA
  # iCFFT is the                    FP = Seq2(FG)*FP
     Channel-based iFFT             # Process amplitude and
  # FI: [N, C, 1, 1]                   phase in Eq.6
  Y = Repeat(FI)                    FI = iCFFT(FA,FP)
  # Repeat FI to the original       # iCFFT is the
     resolution [N, C, H, W]           Channel-based iFFT
                                    # FI: [N, C, 1, 1]
                                    Y = Repeat(FI)
                                    # Repeat FI to the original
  Return Y #[N, C, H, W]               resolution [N, C, H, W]

                                    Return Y #[N, C, H, W]
```

Figure 8: **Pseudo-code of the two variants of the proposed CFTL.** The left is the *Original CFTL* and the right is the *High-oreder CFTL*.

```
def Group CFTL(F):

 # F: input with shape [N, C,
     H, W]
   FG = GlobalPooling(F)
   # Calculation in Eq.5
   #FG: [N, C, 1, 1]
   [FG_1,..FG_K] = Split(FG)
   # Split FG into K groups in
      Eq.10
   # FG_i: [N,C/K,1,1]
   for i in [1, K]:
       FA_i, FP_i = CFFT(FG_i)
       # Applying Channel-based
          FFT in Eq.3 and Eq.4
       # FAi and FPi are the
          derived amplitude and
          phase
       FA_i = Seq1i(FG_i)*FA_i
       FP_i = Seq2i(FG_i)*FP_i
       # Process amplitude and
          phase in Eq.6
       FI_i = iCFFT(FA_i,FP_i)
       # iCFFT is the
          Channel-based iFFT
       # FI_i: [N, C, 1, 1]
   FI = cat([FI_1,..FI_K])
   # Concatention in Eq.11
   Y = Repeat(FI)
   # Repeat FI to the original
      resolution [N, C, H, W]

   Return Y #[N, C, H, W]
```

```
def Spatial-Fourier CFTL(F):

 # F: input with shape [N, C,
     H, W]
   FAS, FPS = FFT(F)
   # Applying Spatial-based
      FFT in Eq.1 and Eq.2
   #FAS, FPS: [N, C, H, W]

   FASA, FASP = CFFT(FAS)
   # Applying Channel-based
      FFT in Eq.3 and Eq.4
   FASA = Seq1(FASA)
   FASP = Seq2(FASP)
   # Process amplitude and
      phase
   FAS =
      Conv_1x1(cat[FASA,FASP])
   # Fuse FASA and FASP as the
      processed amplitude

   FI = iFFT(FSA,FPS)
   # iFFT is the Spatial-based
      iFFT
   # FI: [N, C, H, W]
   Y = FI

   Return Y #[N, C, H, W]
```

Figure 9: **Pseudo-code of another two variants of the proposed CFTL.** The left is the *Group CFTL* and the right is the *Spatial-Fourier CFTL*.

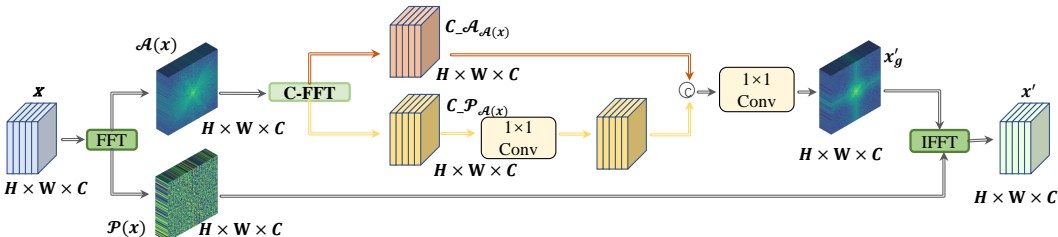

Figure 10: The illustration of Spatial-Fourier CFTL.

instance normalization block as the backbone block. Note that the performance of the CFTL-Net could be further improved if other effective blocks can replace the backbone block.

We train the CFTL-Net on a single GTX3090 GPU with a batch size of 4 and total epochs of 1000. The learning rate is set to $8e^{-4}$ and decays to half every 200 epochs. The loss function is the L1 loss, and the training process is end-to-end. For the training configuration, we train the baseline and the baseline with the CFTL with the same iterations, and both of them are converged for fair comparisons.

**Details of the inverse Fourier transform.** In the main body of Sec. 3.4, we illustrate the operation of the Fourier transform. Here, we also depict the inverse Fourier transform. Given the amplitude component $\mathcal{A}(y)(f)$ and phase component $\mathcal{P}(y)(f)$, the real and image parts of the Fourier

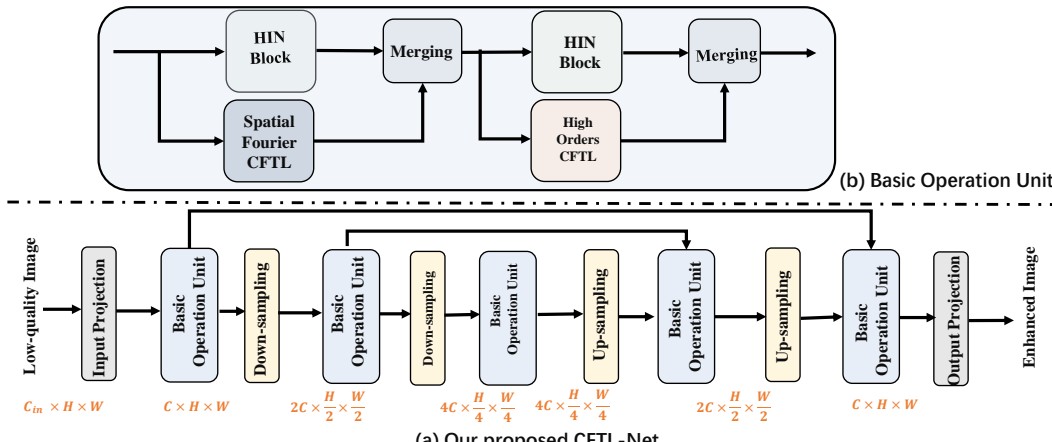

Figure 11: The illustration of the CFTL-Net, which is an encoder-decoder-based architecture.

representation are obtained:

$$
\begin{aligned}
R(y) &= \mathcal{A}(y)(f) \odot cos(\mathcal{P}(y)(f)), \\
I(y) &= \mathcal{A}(y)(f) \odot sin(\mathcal{P}(y)(f)).
\end{aligned}
\tag{13}
$$

Then, $\mathcal{F}(y)(f)$ is formulated by $R(y)$ and $I(y)$, and perform inverse Fourier transform as:

$$
y = \mathcal{F}^{-1}(y)(f) = \frac{1}{C} \sum_{f=0}^{C-1} \mathcal{F}(y)(f) e^{j2\pi \frac{c}{C} f}.
\tag{14}
$$

## B EXTEND CFTL ON OTHER TASKS

**Extension on Image Dehazing.** Following (Dong et al., 2020), we employ the RESIDE dataset (Li et al., 2019a) consisting of Indoor and Outdoor parts for evaluations. We adopt the network of MSBDN (Dong et al., 2020) and FFA-Net (Qin et al., 2020) as the baselines for validation. The results are presented in Table 6.

**Extension on Guided Image Super-resolution.** We apply the original CFTL to the GPPNN (Xu et al., 2021b) and PANNet (Yang et al., 2017) baselines in the pan-sharpening task, which is a common task in guided image super-resolution. We integrate it when fusing pan and multi-spectral features. The experiments are conducted on the WorldView II dataset (Zhou et al., 2022) and the results are shown in Fig. B. The results further suggest the effectiveness of the CFTL.

| Settings | #Param | FLOPs | RESIDE(ITS) | RESIDE(OTS) |
|---|---|---|---|---|
| MSBDN (Baseline) | 31.35M | 166.02G | 29.77/0.9591 | 28.88/0.9581 |
| +Original CFTL | 31.36M | 166.63G | **30.20**/0.9632 | **29.26**/0.9588 |
| +Group CFTL | 31.36M | 166.63G | 29.96/**0.9665** | 29.02/0.9600 |
| +High-order CFTL | 31.36M | 166.63G | 30.13/0.9611 | 29.05/0.9598 |
| +Spatial-Fourier CFTL | 31.36M | 166.76G | 29.89/0.9596 | 29.12/**0.9602** |
| FFA (Baseline) | 4.46M | 1.15T | 36.39/0.9886 | 33.57/0.9840 |
| +Original CFTL | 4.50M | 1.16T | 36.46/0.9902 | 33.89/**0.9912** |
| +Group CFTL | 4.49M | 1.16T | 37.03/0.9936 | **34.28**/0.9901 |
| +High-order CFTL | 4.50M | 1.16T | **37.25/0.9917** | 33.70/0.9842 |
| +Spatial-Fourier CFTL | 4.51M | 1.16T | 36.61/0.9913 | 33.79/0.9904 |
| CFTL-Net | 0.028M | 3.64G | 33.91/0.9829 | 31.62/0.9772 |

Table 6: Results of image dehazing.

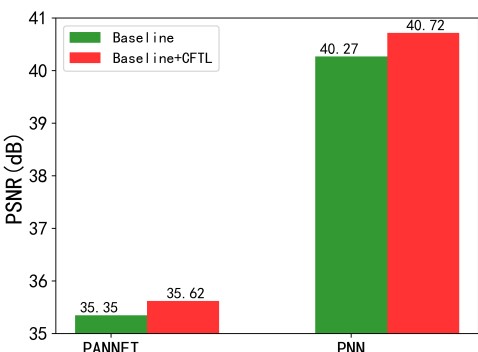

Figure 12: Quantative results of Pan-sharpening.

## C MORE ILLUSTRATION ABOUT GLOBAL INFORMATION FOR IMAGE ENHANCEMENT

In the main body, we describe that the global average pooling equals $\mathcal{A}(0,0)$ in the amplitude. Here, we further verify this from two sides. Typically, the Spatail Fourier transform is expressed as:

$$F(x)(u,v) = \frac{1}{\sqrt{HW}} \sum_{h=0}^{H-1} \sum_{w=0}^{W-1} x(h,w) e^{-j2\pi(\frac{h}{H}u + \frac{w}{W}v)}. \tag{15}$$

The center point of the amplitude spectrum means that u and v are 0. The formula is as follows:

$$F(x)(0,0) = \frac{1}{\sqrt{HW}} \sum_{h=0}^{H-1} \sum_{w=0}^{W-1} x(h,w). \tag{16}$$

It can be seen that the above formula is essentially to find the average value of the entire feature map. Therefore, taking the center point of the amplitude spectrum is equivalent to global average pooling (GAP).

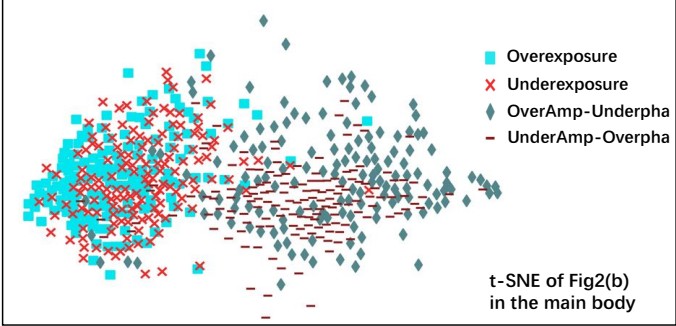

(a) t-SNE of samples in Fig 2 of the main bofy, by swapping A(0,0) in the amplitude, the Underexposure and UnderAmp-Overpha tend to be clustered, while Overexposure and OverAmp-Underpha tend to be clustered, depict A(0,0) contains representative global information. .

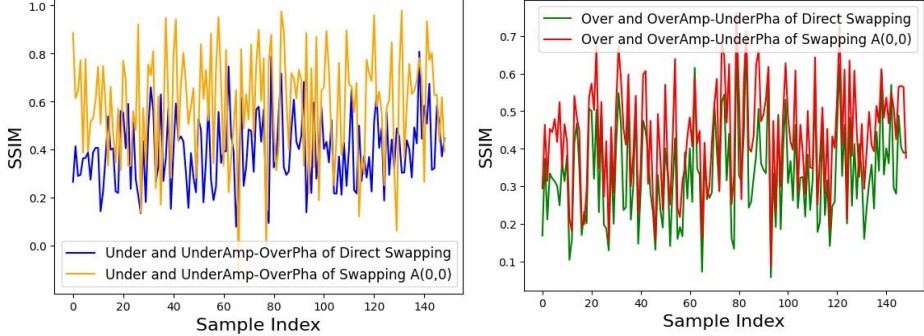

(b) SSIM of the high-frequency between the original sample and swapped sample of two swapping manners, where the high frequency is extracted by substracting the blurred part. Swapping A(0,0) tend to achieve higher SSIM due to its less resulted artifacts.

Figure 13: Different from swapping the amplitude component in (a), we find swapping $\mathcal{A}(0,0)$ in (b) can also swap lightness, while the swapped results contain fewer artifacts.

In Fig. 2 of the main body, we illustrate that $\mathcal{A}(0,0)$ comprises most global information about image enhancement by swapping $\mathcal{A}(0,0)$ of the underexposure and overexposure images, and their lightness becomes similar to the swapped image. The statistic result is presented in Fig. 13 (a), indicating that global pooling is an effective global information format related to image enhancement.

Moreover, this phenomenon also depicts the advantages of our proposed method. As shown in Fig. 2 (a), previous swapping amplitude would cause artifacts in the swapped results, while in Fig. 2 (b), swapping $\mathcal{A}(0,0)$ can alleviate this issue to much extent. We further verify this by comparing SSIM between the original sample and the swapped sample regarding their high-frequency component. As

shown in Fig. 13 (b), swapping $\mathcal{A}(0,0)$ leads to higher SSIM in terms of most samples, depicting there exist fewer artifacts of swapping $\mathcal{A}(0,0)$.

We explain the reasons as follows. This is because there exists a mismatch between the swapped amplitude and the original phase components, which is also referred to as dis-conjugacy in signal processing. Therefore, we argue that previous methods directly processing information in the Fourier domain ignore this issue, and thus may limit the further improvement of performance. Instead, our proposed information processing format avoids this issue while also processing global information that is highly related to image enhancement.

## D   MORE ILLUSTRATIONS AND DISCUSSION ABOUT THE MOTIVATION

**Discussion of the motivation.** The motivation behind this work can be attributed to three aspects. (1) Channel dimension. The channel-dimension relationship reflects the feature information property. For some tasks, such as the style transfer, the Gram matrix can reflect the style information effectively. Therefore, we propose the channel-dimension Fourier transform that provides an alternative format to construct the relationship of channel dimension, which reflects the property like global style information. (2) Fourier transform. Previous methods (i.e., FFC (Chi et al., 2020), GFNet Rao et al. (2021)) have verified the effectiveness of conducting the operation in the Fourier space, which processes the global information conveniently referring to the spectral theory (Chi et al., 2020). Meanwhile, applying the operation in the channel-dimension Fourier space has not been fully explored, which can also affect the channel-dimension information effectively and conveniently. (3) Global information processing. Since the global information is strongly related to image enhancement as illustrated in Fig. 2 in the main paper, the channel-dimension Fourier can enhance it to the high-dimension space, leading to an effective process of the above global information and thus improve image enhancement performance.

**More illustrations of the motivation.** We provide more illustrations of the motivation in Fig. 14. As can be seen, the discriminability of the feature is dependent on the expressiveness of the channel relationship. Therefore, we suspect that enhancing the channel relationship can improve the above discriminability, which can be modeled using the channel-dimension Fourier transform.

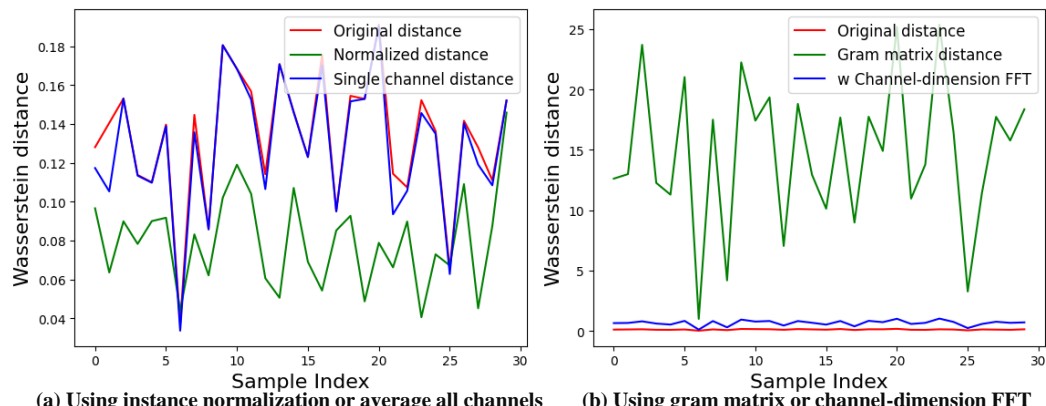

(a) Using instance normalization or average all channels   (b) Using gram matrix or channel-dimension FFT

Figure 14: The distribution distances between underexposure and overexposure using different operations under the toy experiment. The left part depicts that reducing the channel representation ability leads to lower discriminativity, while the right part depicts that exploring the channel relationship using the two operations enhances discriminativity.

## E   MORE DISCUSSION ABOUT TOY EXPERIMENT

We provide more discussions about the toy experiments.

**More illustration of enhancing the discriminability of features using channel-dimension FFT.**

In the supplement of Fig. 4 in the main body, which measures the distribution distance of all samples, we provide the distribution distance of each test sample on the SICE dataset for the toy experiment in Fig. 15. As can be seen, the channel-dimension Fourier transform increases the differences between underexposure and overexposure samples.

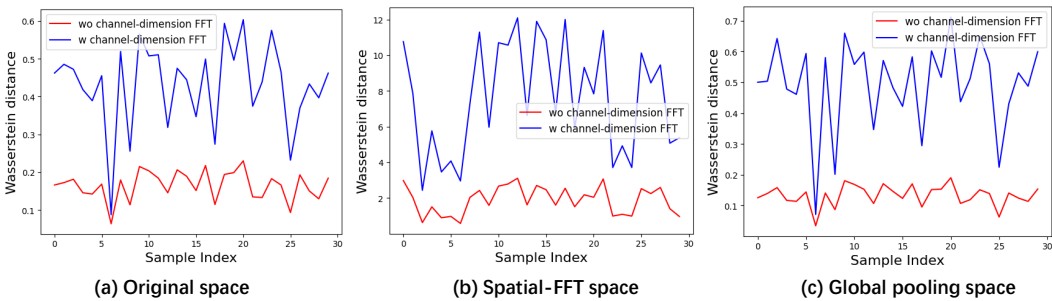

(a) Original space      (b) Spatial-FFT space      (c) Global pooling space

Figure 15: The results of the toy experiment on sample levels in different spaces measure the distribution distances between underexposure and overexposure w/ and w/o using channel-dimension FFT.

**Feature visualization of applying channel-dimension Fourier transform.**

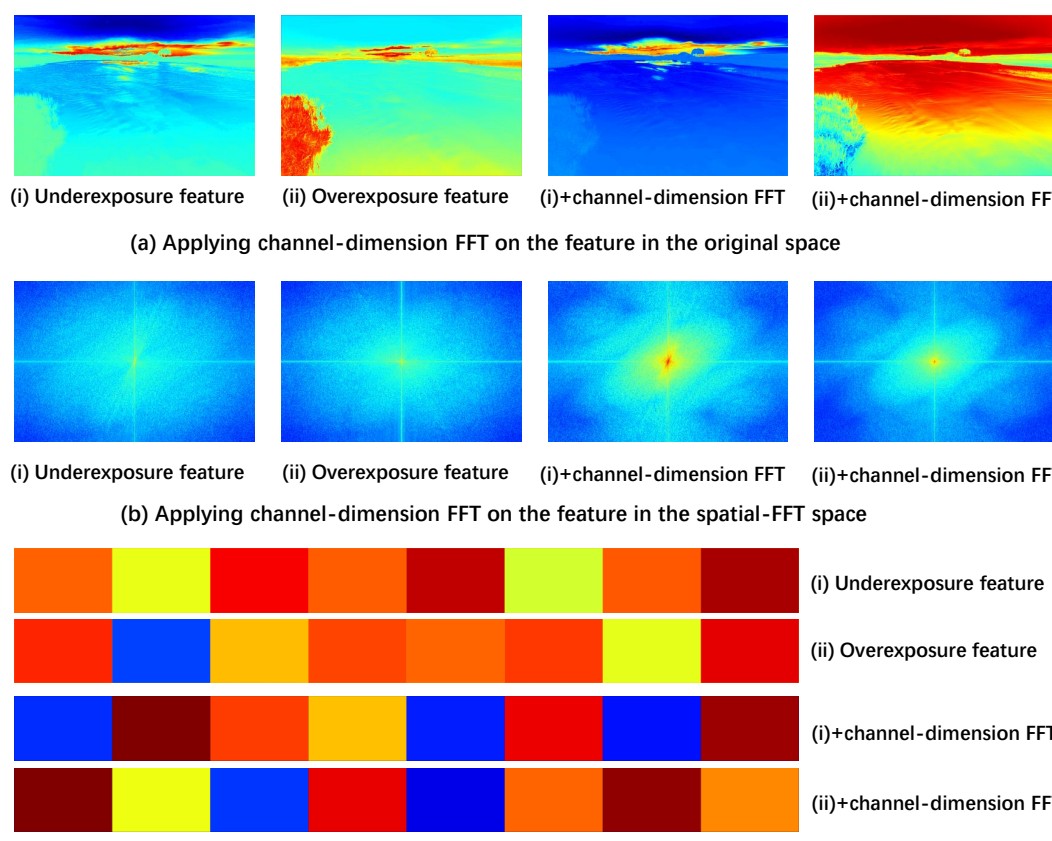

(i) Underexposure feature    (ii) Overexposure feature    (i)+channel-dimension FFT    (ii)+channel-dimension FFT

(a) Applying channel-dimension FFT on the feature in the original space

(i) Underexposure feature    (ii) Overexposure feature    (i)+channel-dimension FFT    (ii)+channel-dimension FFT

(b) Applying channel-dimension FFT on the feature in the spatial-FFT space

(i) Underexposure feature

(ii) Overexposure feature

(i)+channel-dimension FFT

(ii)+channel-dimension FFT

(c) Applying channel-dimension FFT on the feature in the global pooling space

Figure 16: Feature visualization of applying channel-dimension Fourier transform to features in different spaces. Since the global pooling space is difficult to show in one channel, we present all channels of their features.

In supplement to the above statistic results of the toy experiment, we provide feature visualization of applying channel-dimension Fourier transform in different spaces. As shown in Fig. 16, apply-

ing the channel-dimension Fourier transform leads to a more discriminative appearance between underexposure and overexposure features, which corresponds to the previous results.

# F   MORE ILLUSTRATIONS OF THE CFTL'S MECHANISM

**Making the other part focus on learning local representation.** Since the CFTL mainly focuses on learning the global representation, the other part can better capture local representation. We present the feature visualization in Fig. 17, where the features come from the backbone of DRBN trained on the exposure correction task. It can be seen that the introduction of the CFTL brings more texture learning than other comparison operations, depicting its effectiveness.

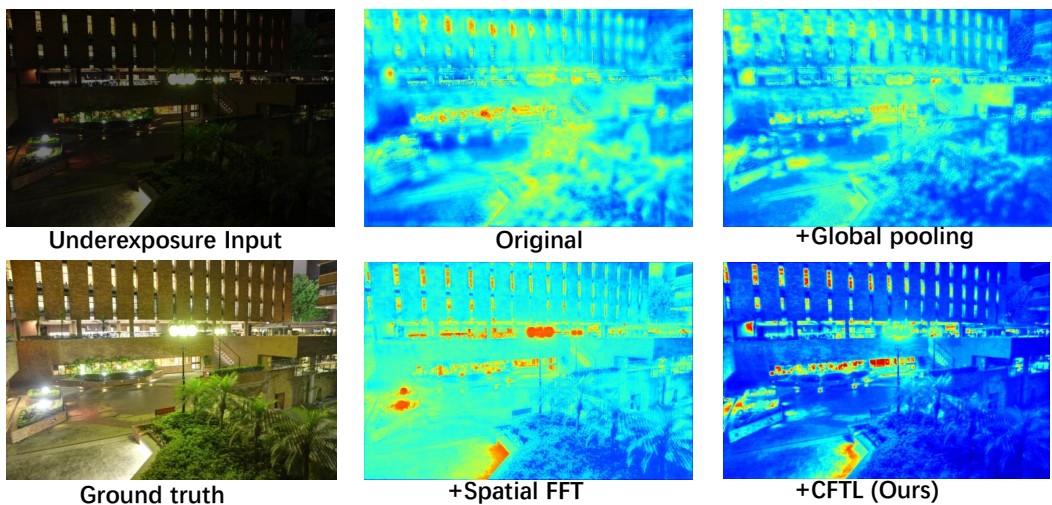

Figure 17: Feature visualization of the local branch on the DRBN-based backbone. The sample is from the SICE dataset. Our proposed CFTL enables the local part to capture more textures.

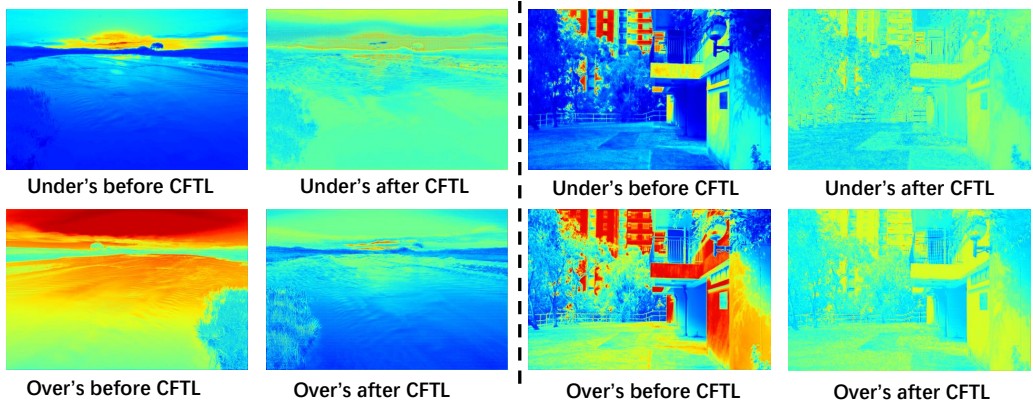

Figure 18: Feature visualization of underexposure and overexposure samples before and after processing by the CFTL. The gap between underexposure and overexposure features is reduced significantly after CFTL.

**Adjust lightness information effectively.**   We present the visualization results of underexposure and overexposure features before and after processing by the CFTL. The features are from the backbone of DRBN trained on the exposure correction task. The results are presented in Fig. 18. As can be seen, the CFTL can effectively reduce the gap between underexposure and overexposure, depicting its effectiveness in adjusting the representation of lightness. Moreover, we compare the distribution distance between underexposure and overexposure in different operations; the CFTL gets the lowest Wasserstein distance of 0.1682, while pooling attention and Sptail Fourier gets 0.2622 and 0.1991, respectively. This result also shows the effectiveness of the proposed CFTL.

## G    MORE DISCUSSION ABOUT THIS WORK

**The reason why CFTL is designed in this format.**    We perform CFTL in the global information representation, where most are based on the global pooling-based representation. The reasons for its design formats are summarized as follows: **(1)** global pooling-based information can represent the most information about global representation. We have detailed its advantages in Sec. C; **(2) one question is why not apply channel-based Fourier transform on the original feature?** We have attempted to do it, but the performance is not good, as shown in Table 5 in the ablation study. Moreover, we have also presented the feature in Fig. 19, and the resulting features appear to have similar properties to the original feature, which would not help improve the learning process. Instead, global pooling information is more suitable for existing convenient operations such as 1x1 convolution to conduct learning. **(3)** We perform operations on the amplitude and phase components because they have explicit meanings in signal processing, and here, they correspond to the information energy distribution and information position distribution along the channel dimension. Besides, the discrepancies between underexposure and overexposure samples of the amplitude component are higher than the original Fourier representation, as shown in Fig. 20, which would contribute to enhancing the discriminability.

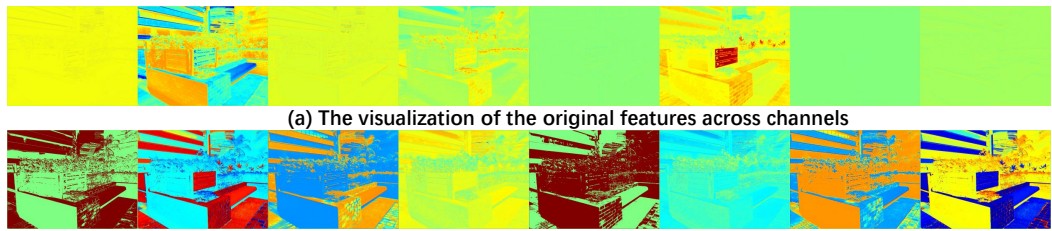

(a) The visualization of the original features across channels

(b) The visualization of the features converted by the channel-dimension Fourier transform directly without global pooling

Figure 19: Feature visualization of applying channel-dimension Fourier transform on the spatial feature (bottom), which also exhibits spatial diversity like original features (top).

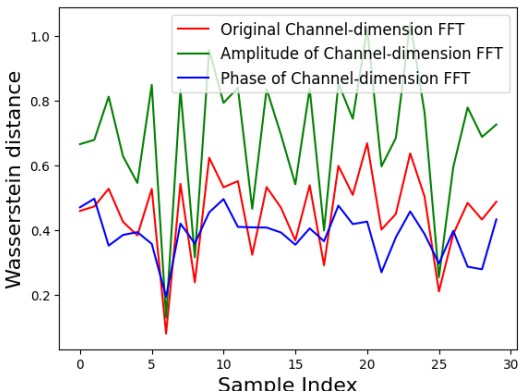

Figure 20: The distribution distances between underexposure and overexposure using different operations under the toy experiment. The amplitude components exhibit higher discrepancies between underexposure and overexposure samples than the original channel-dimension Fourier feature.

**Discussion why "channel-dimension discrepancy" is important for image enhancement.**    (1) The powerful ability of the neural network is to convert the image information to high-dimension with discriminability. The channel-dimension Fourier transform can also enhance the ability to process information in high-dimension. (2) The channel-dimension relationship reflects the discriminative global information property that can contribute to image enhancement, which is similar to the conclusion in (Mustafa et al., 2022), where the polynomial integration of different channels enhances the image enhancement process. (3) The discriminability at the feature level also represents that the sub-sequential filter can respond to different features diversely, which has been proven in (Park et al., 2023), leading to the improvement of image processing.

**Discussion why CFTL is effective.**    Besides the experiments and other sections about the mechanism of why CFTL works, we present other explanations: **(1)** As depicted in the main body, the

channel-based relationship reflects the global style information, such as the Gram matrix. Based on this, the CFTL also learns the relationship of channel information, thus affecting the global style information effectively. **(2)** Applying channel-based Fourier transform derives the global representation of channel information. Therefore, only changing a point in the transformed feature would lead to significant information change in the original channel information, thus leading to the effectiveness of CFTL. **(3)** The proposed mechanism also decouples the learning of global and local information, facilitating the learning of the whole framework. **(4)** Many image enhancement tasks are based on channel prior, such as dark channel prior or rank-one prior, and we believe our designed format also attempts to exact the channel prior in the feature level for image enhancement.

**Some limitations of CFTL.** However, there are also some limitations of CFTL: **(1)** it still occupies some computation costs, which need to be further improved; **(2)** its effects on some popular techniques such as diffusion model, have not explored; **(3)** the more comprehensive experiments on broader Low-level vision tasks (*e.g.*, image de-noising and image de-blurring) have not been explored; **(3)** we mainly apply the CFTL on the lightweight and classic backbones, while some recent backbones with huge parameters have not been explored with the CFTL. **Note that the focus of this work is not to bring a new operation to improve the state-of-the-art, and we aim to introduce a new global information representation format to depict its effectiveness and facilitate the application of classic backbones.**

**Other potential application formats.** According to the above discussions, there are some potential formats for the CFTL: **(1)** conducting attention operations on the channel-based Fourier transformed features without pooling, which could extend the proposed mechanism to the spatial dimension features. **(2)** applying CFTL on the wavelet-based features, which could learn the wavelet coefficients conveniently. **(3)** applying CFTL to fuse features from different architectures or positions, facilitating connecting their channel relationship. **(4)** applying CFTL as a loss function, which could facilitate the optimization of learning global information.

**More potential application for other related tasks.** (1) For tasks such as shadow removal or image harmony, different channels reflect the different properties of the image. For instance, some channels are more related to the shadow regions. Therefore, applying the channel-dimension Fourier transform helps identify and enhance the representation for shadow information processing. (2) For a task such as style transfer, since the channel-dimension Fourier transforms enhances the discriminability of the feature representation, the discriminability about the style can also be strengthened, and thus, it could provide an alternative space to conduct effective style transfer.

## H  MORE DISCUSSIONS AND RESULTS ABOUT THE GENERALIZATION ABILITY

**Whether the proposed method would affect the generalization ability.** (1) The generalization ability of image enhancement is not easily affected by global information. In fact, the mappings between different pairs are quite different in image enhancement tasks due to the inconsistency of the global information mapping (i.e., lightness), but they share a commonality in non-global information components (i.e., texture). Since the generalization ability depends more on learning the commonality component across different images, it would not be easily affected by global information. (2) We also provide the numerical results in Table 7, where we apply the model trained on the LOL dataset to test on the Huawei dataset and vice versa. The results suggest that the generalization performance remains constant with our proposed.

## I  MORE DISCUSSIONS ABOUT DIFFERENT IMPLEMENTATION FORMATS

**The reason why we design different formats.** (1) In terms of performance improvement, the variants of the CFTL can achieve performance improvement in most baselines. However, it is noticed that different formats of the CFTL achieve different performance gains in different baselines. Therefore, we cannot give a certain conclusion about which format can achieve the best performance when the baseline is unknown. The various formats provide alternatives. (2) In terms of extensibility, dif-

| Settings | Trained on LOL/Test on Huawei | Trained on Huawei/Test on LOL |
|---|---|---|
| Restormer (Baseline) | 19.50/0.6407 | 18.39/0.7697 |
| +Pooling attention | 19.25/0.6454 | 18.60/0.7415 |
| +Spaial Fourier | 19.33/0.6436 | 18.48/0.7141 |
| +Original CFTL | 19.20/0.6430 | 18.62/0.7775 |
| +Group CFTL | 19.45/0.6444 | 18.98/0.7806 |
| +High-order CFTL | 19.39/0.6460 | 18.19/0.7689 |
| +Spatial-Fourier CFTL | 19.34/0.6453 | 18.36/0.7716 |

Table 7: Quantative results of evaluating the generalization ability in terms of PSNR/MS-SSIM.

ferent formats provide different views of the channel-dimension Fourier transform. For example, the group CFTL illustrates a balance between the original CFTL and the global pooling with divided groups. Therefore, similar designs, such as wavelet transform and fractional Fourier transform, can also be included for implementation. Meanwhile, the high-order CFTL implies a potential to introduce more abstract information, such as semantic information, into the global vector.

**The relationship between different implementation formats.** For different implementation formats, they share the same principle as "applying the channel-dimension Fourier transform in different spaces". Therefore, the original CFTL is implemented in the global pooling space, while the other formats are implemented in other similar spaces with global information property derived from the global pooling space. We explain their relationships as follows: (1) From the view of global information representation ability, the group CFTL is a degenerate version of the original CFTL with fewer channel information modeling in each group; the high-order CFTL is an evolution version of the original CFTL with more statistics involved; the spatial-Fourier CFTL is another version that expands the representation of global information with more frequencies, but may conquer the issue of unnecessary information as illustrated in Fig. 2 in the main paper. (2) From the view of operation spaces, the high-order CFTL and the group CFTL are all implemented in a single channel dimension of a vector, while the spatial-Fourier CFTL is implemented in spatial spaces with three dimensions, which brings more computation costs. (3) From the view of operation formats, the group CFTL does not involve calculating more statistics, while the other two formats involve calculating more statistics about high-order statistics (high-order CFTL) and frequency statistics (spatial-Fourier CFTL). After all, all of them utilize the 1x1 convolution for their operation.

**The motivation of different design formats.** (1) For the group CFTL, we aim to apply the channel-dimension Fourier transform for the partial channels, which acts as group convolution in the CNNs, which aims to reduce parameters. Moreover, we also implement this format to keep a balance between the original CFTL and the global pooling operation without any channel-dimension Fourier transform, where the group acts as the window in the window Fourier transform and wavelet transform. (2) For the high-order CFTL, we aim to strengthen the capability of the global pooling vector, which introduces more information about the contrast and abstract information in the feature. To this end, the high-order CFTL enhances the feature representation with global intensity information and global contrast information. (3) For the spatial-Fourier CFTL, we expand the global information by introducing more frequencies that extend the global pooling space to the Fourier space. In this way, applying the channel-dimension Fourier transform enhances the global representation across various frequencies.

## J   MORE DETAILS ABOUT EXPERIMENTAL SETTINGS

**Illustration of the comparison operator.** In Fig. 21, we present the illustration of the two comparison operators in the experiments.

**Illustration of how to integrate the CFTL in the backbone.** In Fig. 22, we present how to integrate the CFTL in the backbone. For the backbone with sequential blocks, we usually place the CFTL on the first part of the block. While for the backbone with an encoder-decoder backbone, we place the CFTL on its shallow layers.

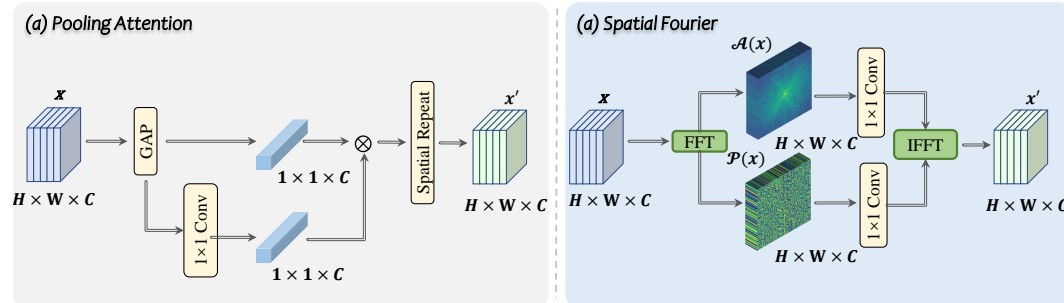

Figure 21: The illustration of the two comparison operators in the experiments.

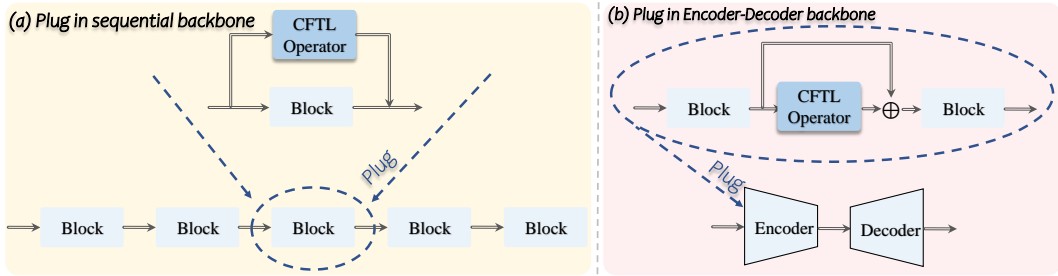

Figure 22: The illustration of how to integrate the CFTL in existing backbones.

## K MORE DISCUSSIONS WITH OTHER ARCHITECTURES AND OPERATIONS

**The relationship with the FcaNet (Qin et al., 2021).** (1) Differences: FcaNet regards the global pooling in the channel attention as a special case of DCT transform. Then, it improves the global pooling by introducing higher-order DCT coefficients. Our method applies the operation for the global pooling vector with Fourier transform instead of expanding the global pooling and could be integrated with FcaNet on its derived vector. (2) Similarity: Both our work and FcaNet aim to construct the channel relationship from the frequency perspective. FcaNet derives the relationship by arranging different group channels with different frequencies, while our method unifies different channels in the frequency space to conjunct all channel information.

**Discussion with other operations such as channel attention and spatial attention.** (1) We argue that some baselines are constructed based on spatial attention or channel attention, such as Restormer, which verifies that our method can improve the performance of more contemporary architectures with this attention. (2) While we validate our method surpasses the global pooling like channel attention, the spatial attention would introduce more computation costs and is different from our proposed channel-dimension modeling. We present the results of adding spatial attention (Woo et al., 2018) to the DRBN network in the LOL dataset in Table 8, which has a lower performance than our proposed method.

| Settings | DRBN (Baseline) | +Spatial Attention | +Original CFTL (Ours) |
|---|---|---|---|
| PSNR/SSIM | 20.73/0.7986 | 21.05/0.8323 | 23.71/0.8492 |

Table 8: Comparison with the spatial attention on the LOL dataset.

**Discussion with the architecture with the large receptive field.** (1) We implement our method mainly on simple and lightweight architectures, which can facilitate the application of simple architectures. Additionally, Restormer with a large receptive field can be improved with our proposed method. (2) The mentioned architecture with a large receptive field enhances the global information with a huge computation burden, while our method is orthogonal to these methods that supplement a lightweight method. (3) Besides the Restormer, we also supplement another architecture SNRformer with a large receptive field as the baseline as illustrated in Table 12, the performance is also improved.

## L  MORE ABLATION STUDIES FOR INVESTIGATING THE CFTL

Referring to the main body, we also perform ablation studies in the DRBN backbone on the exposure correction task (SICE dataset).

Firstly, we discuss why we perform the ablation studies on the exposure correction task in two folds: (1) The lightness among the different input images is quite different, which can be well-represented by our proposed CFTL mechanism. (2) The lightness adjustment directions of different exposures are quite different in this task. Therefore, this is a challenging task that can prominently evaluate how the proposed components contribute to performance improvement.

**Ablation study of different orders in High-order CFTL.** Defautly, we set $k$ in High-order CFTL to 2. Here, we also present other combinations of different orders' results in Table 9. As can be seen, most of them achieve comparable performance with the default setting but outperform the original CFTL. These results demonstrate the reasonableness of the default setting and also the effectiveness of the High-order CFTL.

| Settings (order) | Baseline (DRBN) | +CFTL | +CFTL (1+2) | +CFTL (1+4) | +CFTL (1+2+4) |
|---|---|---|---|---|---|
| PSNR/MS-SSIM | 17.65/0.6798 | 21.32/0.7250 | **21.64**/ 0.7243 | 21.53/ 0.7247 | 21.56/ **0.7262** |

Table 9: Investigating different orders (denoted as 1+k) in High-order CFTL on the SICE dataset for exposure correction, where "+CFTL (1+2)" is the default High-order CFTL in the manuscript.

**Ablation study of different groups in Group CFTL.** Defautly, we set $K$ in Group CFTL to 4. Here, we also present setting other group numbers' results in Table 10. It can be seen that setting the group numbers to 1, 2, and 4 has similar results, while increasing the number to 8 would lead to a performance drop. This result suggests the reasonableness of the default setting, where setting group 4 achieves comparable performance with introducing fewer parameters.

| Settings (order) | Baseline (DRBN) | +CFTL (K=1) | +CFTL (K=2) | +CFTL (K=4) | +CFTL (K=8) |
|---|---|---|---|---|---|
| PSNR/MS-SSIM | 17.65/0.6798 | 21.32/0.7250 | 21.26/0.7209 | 21.30/0.7177 | 20.86/0.7093 |
| Parameters | 0.532M | 0.534M | 0.533M | 0.532M | 0.532M |

Table 10: Investigating the number of groups $K$ in Group CFTL on the SICE dataset for exposure correction, where "+CFTL(K=1)" is the default Group CFTL in the manuscript.

**Ablation study of different settings in Spatial-Fourier CFTL.** In Sec. A, we present the details of the Spatial-Fourier CFTL. In Table 11, we present other configurations of the Spatial-Fourier CFTL. As can be seen, applying CiFFT (channel-based inverse Fourier transform ) would lead to unstable training and results in a "NAN" problem, while processing both amplitude and phase components derived from the spatial-based Fourier transform brings performance drop. Overall, the final setting of the Spatial-Fourier CFTL achieves the best performance.

| Settings | Baseline (DRBN) | +Spatial-Fourier CFTL | +Spatial-Fourier CFTL (+CiFFT) | (a) |
|---|---|---|---|---|
| PSNR/MS-SSIM | 17.65/0.6798 | 21.33/0.7201 | NAN/NAN | 21.20/0.7159 |
| Parameters | 0.532M | 0.536M | 0.536M | 0.537M |

Table 11: Investigating the configuration of Spatial-Fourier CFTL on the SICE dataset for exposure correction, where "(a)" denotes processing both amplitude and phase components derived from the spatial-based Fourier transform.

**Ablation study of investigating the channel number in the CFTL-Net.** We set the channel number of CFTL-Net as 8 in the manuscript. We also investigate setting other channel numbers, and results conducted on the exposure correction and low-light image enhancement are presented in Fig. 23. It can be seen that increasing the channel number would lead to more performance improvement in exposure correction than low-light image enhancement while reducing the channel

number results in a significant performance drop. This could be attributed to the fact that exposure correction requires a stronger ability to adjust different lightness. The results suggest the potential extensive ability of the CFTL-Net with more channel numbers.

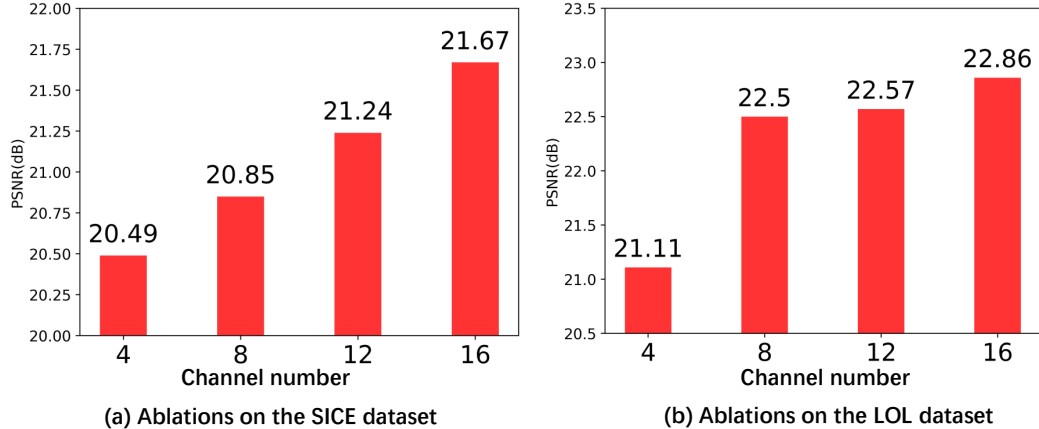

(a) Ablations on the SICE dataset        (b) Ablations on the LOL dataset

Figure 23: Abaltion studies for investigating different channel numbers of the CFTL-Net in exposure correction (left) and low-light image enhancement (right).

## M  MORE RESULTS OF APPLYING THE CFTL IN OTHER BACKBONES

In the main body, we adopt a few networks as the backbone to integrate the CFTL. Here, we employ more networks as the backbone to demonstrate the scalability and effectiveness of the CFTL, which are presented in Table 12, Table 13, Table 14 and Table 15.

**For Low-light image enhancement.** We further employ the Bread (Guo & Hu, 2023) and SNR-former (Xu et al., 2022) as the backbone and perform the experiments on the LOL dataset (Wei et al., 2018). We retrain these networks under the same setting until they are converged for a fair comparison. The extensive results presented in Table 12 validate the effectiveness of the proposed CFTL.

| Settings | Baseline (Bread) | +Spatial Fourier | +Attention Pooling | +Original CFTL | +Group CFTL | +Spatial-Fourier CFTL |
|---|---|---|---|---|---|---|
| PSNR/SSIM | 22.96/0.8383 | 23.22/0.8399 | 23.16/0.8407 | 23.31/0.8403 | 23.37/**0.8414** | **23.42**/0.8409 |
| Settings | Baseline (SNRformer) | +Spatial Fourier | +Attention Pooling | +Original CFTL | +Group CFTL | +Spatial-Fourier CFTL |
| PSNR/SSIM | 23.61/**0.8440** | 23.37/0.8375 | 22.47/0.8281 | **23.72**/0.8371 | 22.99/0.8324 | 23.49/0.8386 |

Table 12: More Comparison over low-light image enhancement on the LOL dataset.

**For exposure correction.** Moreover, we adopt FECNet (Huang et al., 2022) as the backbone for exposure correction. We perform the experiments in the SICE dataset, and the experimental results in Table 13 demonstrate the effectiveness of our method. Note that FECNet is based on the design of spatial Fourier transform as the basic unit. This result shows that our method is compatible with the existing spatial Fourier transform design.

| Settings | Baseline (FECNet) | +Spatial Fourier | +Attention Pooling | +Original CFTL |
|---|---|---|---|---|
| PSNR/SSIM | 20.96/0.6849 | 21.06/0.6913 | 21.14/0.6955 | 21.60/0.7302 |

Table 13: More Comparison overexposure correction on the SICE dataset.

**For SDR2HDR translation.** Moreover, we adopt the first stage of the AGCM (Chen et al., 2021b) as the backbone for SDR2HDR translation. We perform the experiments in the HDRTV dataset (Chen et al., 2021b), and the experimental results in Table 14 demonstrate the effectiveness of our method.

**For underwater image enhancement.** Moreover, we adopt Five A plus Jiang et al. (2023) as the backbone for underwater image enhancement. We perform the experiments in the UIEB dataset (Li et al., 2019b), and the experimental results in Table 15 demonstrate the effectiveness of our method.

| Settings | Baseline (AGCM) | +Spatial Fourier | +Attention Pooling | +Original CFTL |
|---|---|---|---|---|
| PSNR/SSIM | 36.53/0.9624 | 36.58/0.9635 | 36.79/0.9655 | 36.83/0.9657 |

Table 14: More comparisons over SDR2HDR translation on the HDRTV dataset.

| Settings | Baseline (Five A plus) | +Spatial Fourier | +Attention Pooling | +Original CFTL | +Group CFTL | +Spatial-Fourier CFTL |
|---|---|---|---|---|---|---|
| PSNR/SSIM | 23.63/0.9138 | 23.96/0.9157 | 23.79/0.9134 | **24.24/0.9172** | 24.07/0.9157 | 23.98/0.9145 |

Table 15: More Comparison over underwater image enhancement on the UIEB dataset.

## N  MORE RESULTS BY COMPARING WITH MORE METHODS

We provide more quantitative results of the CFTL-Net and other comparison methods on different image enhancement tasks, including low-light image enhancement, exposure correction, and SDR2HDR translation. The results are presented in Table 16 (low-light image enhancement), Table 17 (low-light image enhancement), Table 18 (exposure correction), and Table 19 (SDR2HDR translation). This demonstrates the CFTL-Net achieves an elegant balance between performance and efficiency. Note that the improved version of the method in the main body also performs better than most comparison methods in these tables.

| Method | LOL | | Huawei | | # Param | GFLOPs |
|---|---|---|---|---|---|---|
| | PSNR | SSIM | PSNR | SSIM | | |
| SRIE (Fu et al., 2016) | 12.28 | 0.5962 | 13.04 | 0.4770 | - | - |
| RobustRetinex (Li et al., 2018) | 13.88 | 0.6643 | 14.60 | 0.5593 | - | - |
| RetinexNet (Wei et al., 2018) | 16.77 | 0.4257 | 16.65 | 0.4857 | 0.84M | 148.54 |
| MBLLEN (Lv et al., 2018) | 17.56 | 0.7293 | 16.63 | 0.5264 | 0.45M | 21.37 |
| EnGAN (Jiang et al., 2021) | 17.48 | 0.6746 | 17.03 | 0.5140 | 8.37M | 72.61 |
| GLADNet (Wang et al., 2018) | 19.72 | 0.6802 | 17.76 | 0.5214 | 1.13M | 275.32 |
| Xu et al. (Xu et al., 2020) | 16.78 | 0.7665 | 16.12 | 0.5862 | 8.62M | 68.45 |
| TBEFN (Lu & Zhang, 2020) | 17.35 | 0.7817 | 16.88 | 0.5759 | 0.49M | 24.11 |
| KIND (Zhang et al., 2019) | 20.86 | 0.8023 | 16.48 | 0.5406 | 8.54M | 36.57 |
| ZeroDCE (Guo et al., 2020) | 15.29 | 0.5182 | 12.46 | 0.4074 | 0.08M | 20.24 |
| DRBN (Yang et al., 2020) | 20.13 | 0.8011 | 19.93 | 0.6810 | 0.53M | 42.41 |
| RUAS (Liu et al., 2021) | 16.41 | 0.5004 | 13.76 | 0.5167 | 0.003M | 0.86 |
| KIND++ (Zhang et al., 2021) | 21.30 | 0.8221 | 15.78 | 0.4523 | 8.28M | 2970.50 |
| URetinex (Wu et al., 2022) | 21.32 | **0.8358** | 18.79 | 0.6078 | 1.23M | 68.37 |
| LA-Net (Yang et al., 2023) | 21.71 | 0.8149 | 18.15 | 0.5941 | 0.55M | 185.79 |
| CFTL-Net (Ours) | **22.50** | 0.8139 | **20.91** | **0.6941** | 0.028M | 3.64 |

Table 16: Quantitative results of different methods on the LOL and Huawei datasets for low-light image enhancement. #Param denotes the parameter number. The best and second results are highlighted in **bold** and underline, respectively.

## O  MORE QUALITATIVE RESULTS

Due to the page limit of the main body, we provide more visualization results here. In the main body, we have presented the visual results of exposure correction in Fig. 6. Here, we further respectively present the results of low-light image enhancement (Fig. 24, Fig. 25 and Fig. 26), exposure correction (Fig. 27 and Fig. 28), SDR2HDR translation (Fig. 29), and underwater image enhancement (Fig. 30) as follows. As can be seen, our CFTL can help enhance more correct lightness and color or reduce the structure artifacts.

| Method | PSNR | SSIM | #Param |
|---|---|---|---|
| Whitebox (Hu et al., 2018) | 18.59 | 0.7973 | 8.17M |
| Distort (Park et al., 2018) | 19.54 | 0.7998 | 247.25M |
| HDRNet (Gharbi et al., 2017) | 22.65 | 0.8802 | 0.46M |
| SID (Chen et al., 2018) | 21.49 | 0.8425 | 7.40M |
| DUPE (Wang et al., 2019) | 20.22 | 0.8287 | 0.95M |
| DeepLPF (Moran et al., 2020) | 23.21 | 0.8863 | 0.80M |
| DRBN (Yang et al., 2020) | 22.11 | 0.8684 | 0.53M |
| CSRNet (He et al., 2020) | 23.69 | 0.8981 | 0.034M |
| LA-Net (Yang et al., 2023) | 19.94 | 0.8057 | 0.55M |
| DSN (Zhao et al., 2021) | 23.84 | **0.9002** | 4.42M |
| CFTL-Net (Ours) | **24.03** | 0.8904 | 0.028M |

Table 17: Quantitative results of different methods on the MIT-FiveK dataset for low-light image enhancement. #Param denotes the parameter number. The best and second results are highlighted in **bold** and underline, respectively.

| Method | MSEC | | | | | | SICE | | | | | | #Param | GFLOPs |
|---|---|---|---|---|---|---|---|---|---|---|---|---|---|---|
| | Underexposure | | Overexposure | | Average | | Underexposure | | Overexposure | | Average | | | |
| | PSNR | SSIM | PSNR | SSIM | PSNR | SSIM | PSNR | SSIM | PSNR | SSIM | PSNR | SSIM | | |
| HE (Pizer et al., 1987) | 16.52 | 0.6918 | 16.53 | 0.6991 | 16.53 | 0.6959 | 14.69 | 0.5651 | 12.87 | 0.4991 | 13.78 | 0.5376 | - | - |
| CLAHE (Reza, 2004) | 16.77 | 0.6211 | 14.45 | 0.5842 | 15.38 | 0.5990 | 12.69 | 0.5037 | 10.21 | 0.4847 | 11.45 | 0.4942 | - | - |
| RetinexNet (Wei et al., 2018) | 12.13 | 0.6209 | 10.47 | 0.5953 | 11.14 | 0.6048 | 12.94 | 0.5171 | 12.87 | 0.5252 | 12.90 | 0.5212 | 0.84M | 148.54 |
| DPED (Ignatov et al., 2017) | 20.06 | 0.6826 | 13.14 | 0.5812 | 15.91 | 0.6219 | 16.83 | 0.6133 | 7.99 | 0.4300 | 12.41 | 0.5217 | 0.39M | 94.64 |
| SID (Chen et al., 2018) | 19.37 | 0.8103 | 18.83 | 0.8055 | 19.04 | 0.8074 | 19.51 | 0.6635 | 16.79 | 0.6444 | 18.15 | 0.6540 | 7.40M | 53.12 |
| URetinexNet (Wu et al., 2022) | 13.85 | 0.7371 | 9.81 | 0.6733 | 11.42 | 0.6988 | 17.39 | 0.6448 | 7.40 | 0.4543 | 12.40 | 0.5496 | 1.32M | 68.37 |
| Zero-DCE (Guo et al., 2020) | 14.55 | 0.5887 | 10.40 | 0.5142 | 12.06 | 0.5441 | 16.92 | 0.6330 | 7.11 | 0.4292 | 12.02 | 0.5311 | 0.079M | 20.24 |
| li2021learning (Li et al., 2021) | 13.82 | 0.5887 | 9.74 | 0.5142 | 11.37 | 0.5583 | 11.93 | 0.4755 | 6.88 | 0.4088 | 9.41 | 0.4422 | 0.010M | 0.17 |
| RUAS (Liu et al., 2021) | 13.43 | 0.6807 | 6.39 | 0.4655 | 9.20 | 0.5515 | 16.63 | 0.5589 | 4.54 | 0.3196 | 10.59 | 0.4393 | 0.0014M | 0.86 |
| DRBN (Yang et al., 2020) | 19.74 | 0.8290 | 19.37 | 0.8321 | 19.52 | 0.8309 | 17.96 | 0.6767 | 17.33 | 0.6828 | 17.65 | 0.6798 | 0.53M | 42.41 |
| MSEC (Afifi et al., 2021) | 20.52 | 0.8129 | 19.79 | 0.8156 | 20.08 | 0.8210 | 19.62 | 0.6512 | 17.59 | 0.6560 | 18.58 | 0.6536 | 7.04M | 35.87 |
| CMEC (Nsamp et al., 2021) | 22.23 | 0.8140 | 22.75 | 0.8336 | 22.54 | 0.8257 | 17.68 | 0.6592 | 18.17 | 0.6811 | 17.93 | 0.6702 | 5.40M | 35.71 |
| LA-Net (Yang et al., 2023) | 21.84 | 0.8264 | 21.02 | 0.8164 | 21.35 | 0.8207 | 19.22 | 0.6514 | 17.65 | 0.6025 | 18.44 | 0.6270 | 0.55M | 185.79 |
| LCDPNet (Wang et al., 2022) | 22.35 | 0.8650 | 22.17 | 0.8476 | 22.30 | 0.8552 | 20.71 | 0.6822 | 20.21 | 0.6863 | 20.46 | 0.6843 | 0.96M | 9.40 |
| CFTL-Net (Ours) | **23.04** | **0.8612** | **22.63** | **0.8567** | **22.88** | **0.8594** | **22.26** | **0.6927** | **20.71** | **0.7071** | **21.24** | **0.6999** | 0.028M | 3.61 |

Table 18: Quantitative results of different methods on the MSEC and SICE datasets for exposure correction. #Param denotes the parameter number. The best and second results are highlighted in **bold**.

| Method | PSNR | SSIM | #Param |
|---|---|---|---|
| HuoPhyEO (Huo et al., 2014) | 25.90 | 0.9296 | - |
| Pixel2Pixel (Isola et al., 2017) | 25.80 | 0.8777 | 11.38M |
| CycleGAN (Zhu et al., 2017) | 21.33 | 0.8496 | 11.38M |
| HDRNet (Gharbi et al., 2017) | 35.73 | 0.9664 | 0.46M |
| JSI-GAN (Kim et al., 2020) | 37.01 | **0.9694** | 1.06M |
| Ada-3DLUT (Zeng et al., 2020) | 36.22 | 0.9658 | 0.59M |
| DRBN (Yang et al., 2020) | 36.44 | 0.9671 | 0.53M |
| CSRNet (He et al., 2020) | 35.34 | 0.9625 | 0.034M |
| LA-Net (Yang et al., 2023) | 31.52 | 0.9427 | 0.55M |
| AGCM (Plus) (Chen et al., 2021b) | 36.88 | 0.9655 | 0.034M |
| CFTL-Net (Ours) | **37.37** | 0.9683 | 0.028M |

Table 19: Quantitative results of different methods on the HDRTV dataset for SDR2HDR translation. #Param denotes the parameter number. The best and second results are highlighted in **bold** and underline, respectively.

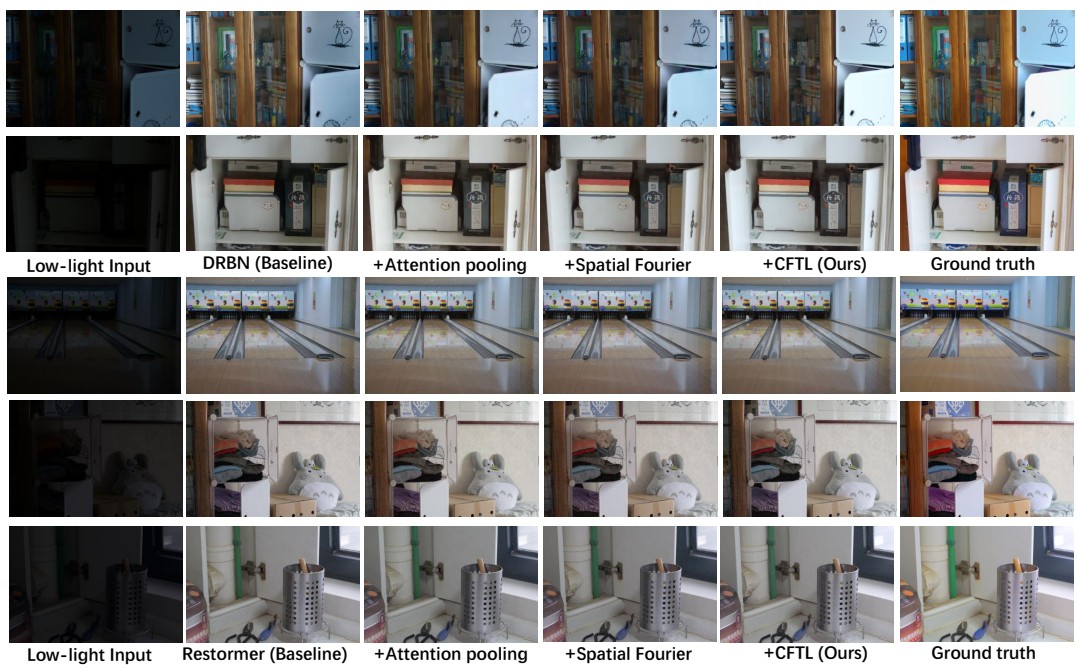

Figure 24: The visualization results on the LOL dataset for low-light image enhancement.

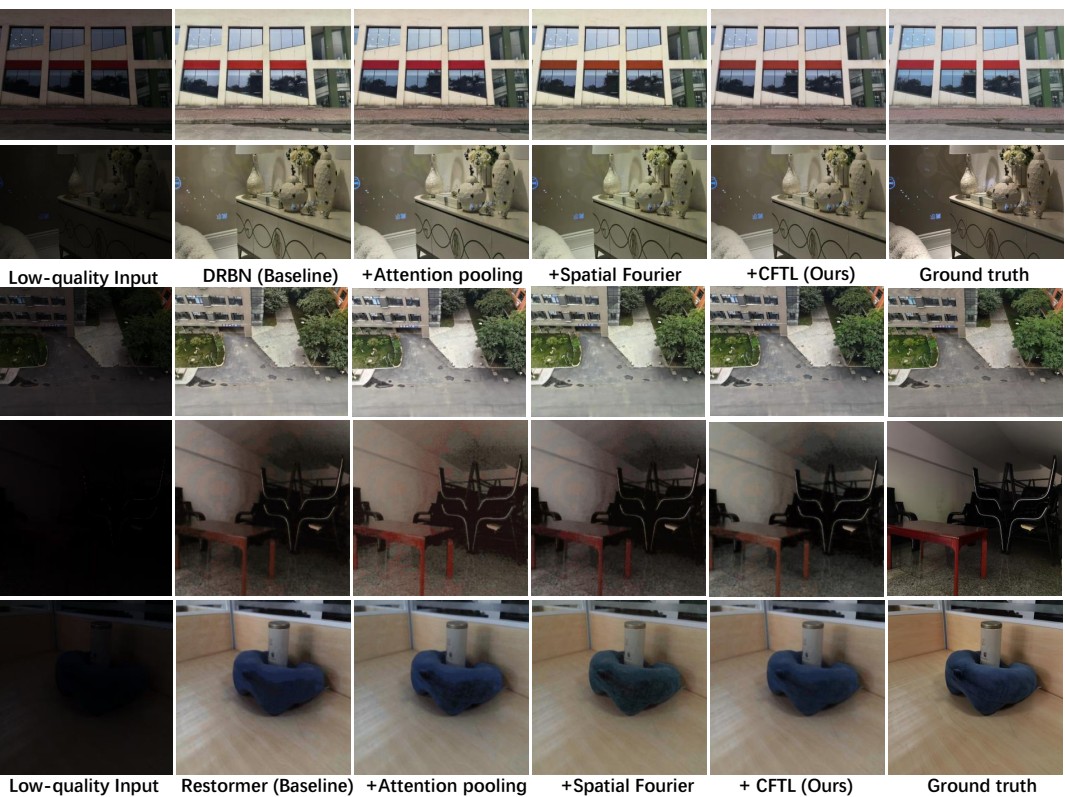

Figure 25: The visualization results on the Huawei dataset for low-light image enhancement.

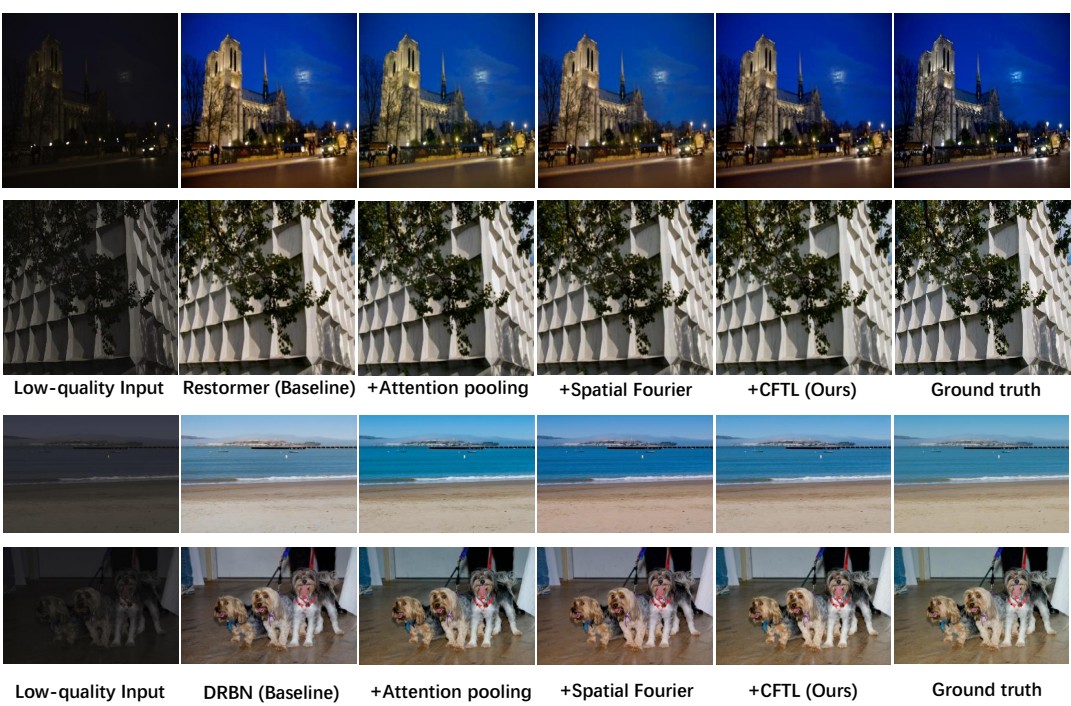

Figure 26: The visualization results on the FiveK dataset for low-light image enhancement.

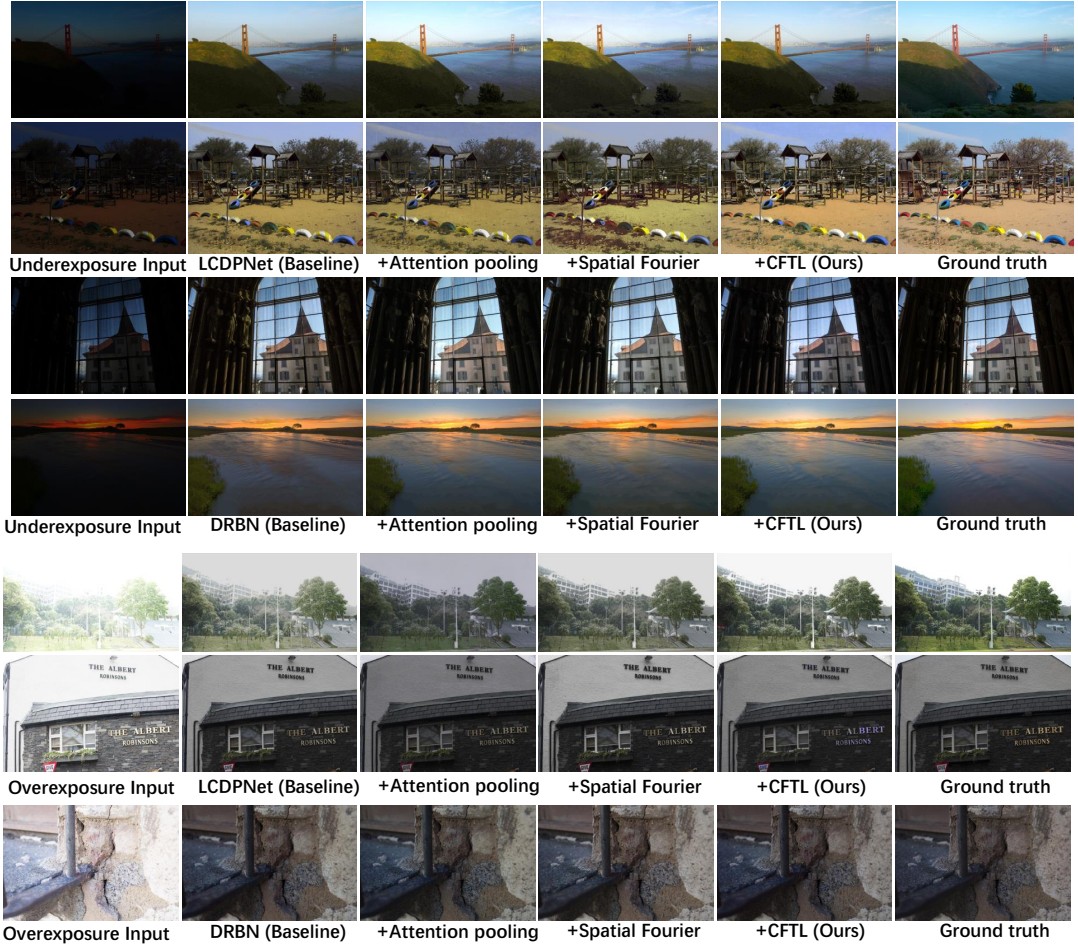

Figure 27: The visualization results on the SICE dataset for exposure correction (top: underexposure correction, bottom: overexposure correction).

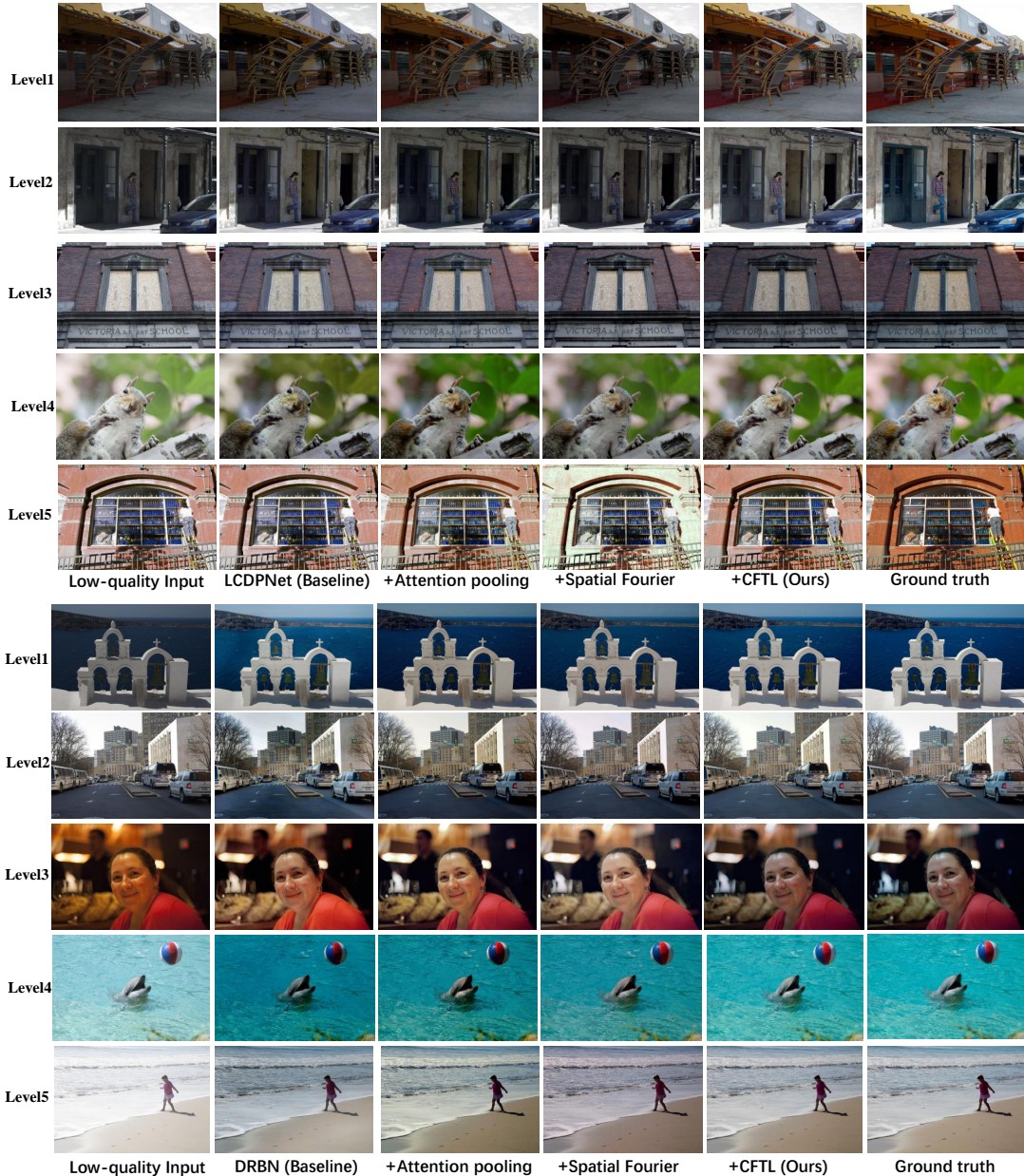

Figure 28: The visualization results on the MSEC dataset for exposure correction with different exposure levels (from level 1 to level 5).

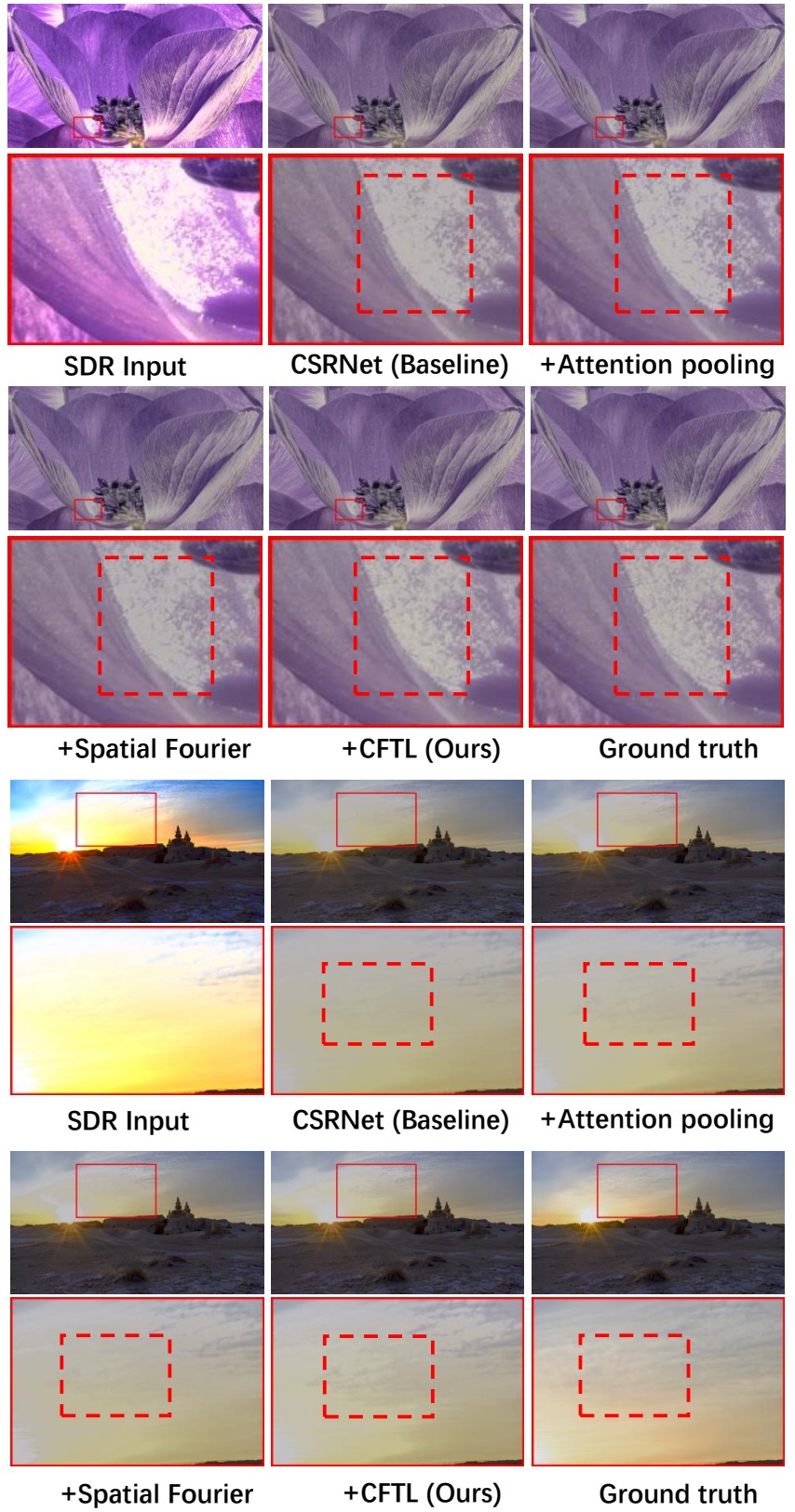

Figure 29: The visualization results on the HDRTV dataset for SDR2HDR translation.

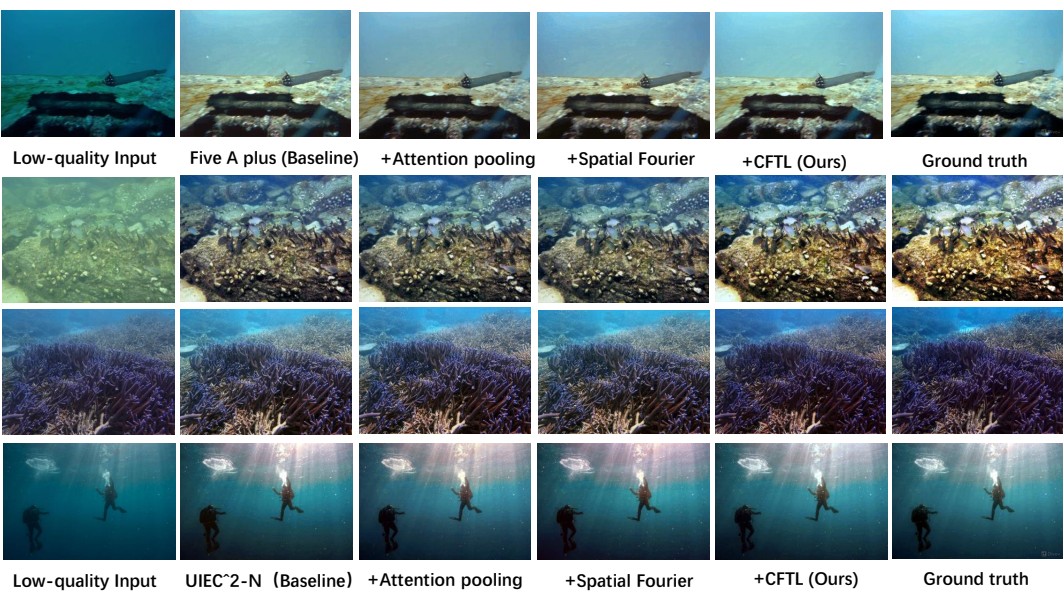

Figure 30: The visualization results on the UIEB dataset for underwater image enhancement.

