# OpenReview forum: "Revitalizing Channel-dimension Fourier Transform for Image Enhancement"
_ICLR.cc/2024/Conference — Submitted to ICLR 2024_

### Official Review · Reviewer_MiY2 · 2023-10-29

**Soundness:** 4 excellent
**Presentation:** 3 good
**Contribution:** 3 good
**Rating:** 8
**Confidence:** 4

**Summary:**

This paper investigate a new mechanism of the channel-dimension Fourier transform to enhance the discriminativity of the global information. Based on the core mechanism, the method design consists of several implementation formats. Extensive experiments over several image enhancement tasks and datasets demonstrates the effectiveness and versatility of the proposed method for image enhancement.

**Strengths:**

1. This paper investigates a new mechanism of channel-dimension Fourier transform, which has not been discussed before as I know. As a new technique, its advantages and mechnism are well-been illustrated.
2. This paper presents quantities of experiments to verify the effectiveness and general ability of the proposed method in both main body and appendix. Moreover, this paper also present quantities of analysis in appendix to exhibit the method’s mechanism, which is inspiring.
3. This paper also provides several implementation formats, which are easy to implement and easy to follow.

**Weaknesses:**

1. This paper contains few visual results. As an image processing paper, the number of visual results are limited to some extent.
2. This paper presents several implementation formats. However, the relationship between these formats are not provided. How to derive different implementation formats from the original one?
3. The ablation studies are mainly conducted on exposure correction as shown in Table 5. However, as a main image enhancement task, this paper need to supplement ablation studies on the low-light image enhancement task to verify the effectiveness of the method’s design.

**Questions:**

Can authors provide more visual results? This is important to depict the effectiveness of the proposed method. Moreover, can authors discuss why the several implementation formats are designed in such manner?

---

> ### Author Response · Authors · 2023-11-19
>
> Thanks for your efforts and constructive suggestions, and the rebuttal is listed as follows.
>
> **Q1 More visual results should be provided.**
>
> **A1:** As suggested, we have provided more results in the appendix, including visual results of low-light image enhancement (Fig. 24， Fig. 25, and Fix. 26), visual results of exposure correction (Fig. 27 and Fig.28), visual results of SDR2HDR on the HDRTV dataset (Fig. 29) and visual results of underwater image enhancement dataset on the UIEB dataset (Fig. 30).
>
> **Q2 What is the relationship between different implementation formats? How are they derived from the original one?**
>
> **A2:** For different implementation formats, they share the same principle as “applying the channel-dimension Fourier transform in different spaces”. Therefore, the original CFTL is implemented in the global pooling space, while the other formats are implemented in other similar spaces with global information property derived from the global pooling space. We explain their relationships as follows: 1) From the view of global information representation ability; the group CFTL is a degenerate version of the original CFTL with fewer channel information modeling in each group; the high-order CFTL is an envoluted version of the original CFTL with more statistics involved; the spatial-Fourier CFTL is another version that expands the representation of global information with more frequencies, but may conquer the issue of unnecessary information as illustrated in Fig.2 in the main paper.  2) From the view of operation spaces, the high-order CFTL and the group CFTL are all implemented in a single channel dimension of a vector; while the spatial-Fourier CFTL is implemented in spatial spaces with three dimensions, which brings more computation costs. 3) From the view of operation formats, the group CFTL does not involve calculating more statistics, while the other two formats involve calculating more statistics about high-order statistics (high-order CFTL) and frequency statistics (spatial-Fourier CFTL). After all, all of them utilize the 1x1 convolution for their operation. We add the above description to make them more clear in Sec.I in the appendix.
>
> **Q3 Why different implementation formats are proposed and implemented in such a manner?**
>
> **A3:** For different implementation formats, we design them based on different reasons. 1) For the group CFTL, we aim to apply the channel-dimension Fourier transform for the partial channels, which acts as group convolution in the CNNs, which aims to reduces parameters. Moreover, we also implement this format to keep a balance between the original CFTL and the global pooling operation without any channel-dimension Fourier transform, where the group acts as the window in the window Fourier transform and wavelet transform. 2) For the high-order CFTL, we aim to strengthen the capability of the global pooling vector, which introduces more information about the contrast and abstract information in the feature. To this end, the high-order CFTL enhances the feature representation with global intensity information and global contrast information. 3) For the spatial-Fourier CFTL, we expand the global information by introducing more frequencies that extend the global pooling space to the Fourier space. In this way, applying the channel-dimension Fourier transform enhances the global representation across various frequencies. We add the above description to make them more clear in Sec.I in the appendix.
>
> **Q4 Ablation studies on more datasets and tasks should be provided.**
>
> **A4:** Thank you for your kind suggestions. The reason why we perform ablation studies on exposure correction can be explained in two folds: 1) The lightness among the different input images is quite different, which can be well-represented by our proposed CFTL mechanism. 2) The lightness adjustment directions of different exposures are quite different in this task. Therefore, this is a challenging task that can prominently evaluate how the proposed components contribute to performance improvement.
>
> | Configurations                    | PSNR/SSIM        |
> |----------------------------------|---------------------|
> | Baseline (DRBN)                     | 20.73/0.7986        |
> | +CFTL                        |   **23.71/0.8492**    |
> | +CFTL w/o global pooling               | 19.91/0.8317        |
> | +CFTL w/o channel-based ifft         | 22.66/0.8453        |
> | +CFTL w/o processing amplitude         | 22.78/0.8438        |
> | +CFTL w/o processing phase            | 23.29/0.8445        |
>
> Additionally, due to time constraints, we provide more ablation studies on more datasets such as the low-light image enhancement task on the LOL dataset. We have included the corresponding results in Table 5， and include the above discussion in the appendix, Sec. L.

---

### Official Review · Reviewer_Tiry · 2023-10-29

**Soundness:** 3 good
**Presentation:** 3 good
**Contribution:** 3 good
**Rating:** 10
**Confidence:** 5

**Summary:**

This paper proposes a new Fourier transform for image enhancement, consisting of three steps: applying Fourier transform to the channel dimension to obtain channel-wise Fourier domain features, performing a channel-wise transformation on both its amplitude and phase components, and then reverting back to the spatial domain. Based on these three steps, three strategies of channel transform are designed. Extensive experiments are conducted to show the effectiveness of the proposed method.

**Strengths:**

1.	The proposed method provides a perspective to formulate new Fourier transform for image enhancement, and has achieved SOTA performance on different tasks with efficient parameter number and flops.

2.	The experiments are sufficient in both main paper and supp.

3.	The writing and organization of this paper is great.

**Weaknesses:**

1.	The ablation studies in Table 5 can be conducted on more datasets to comprehensively show the effects of each component.

2.	More visual comparisons should be provided as these of Fig. 6.

**Questions:**

Is the baseline and the baseline with CFTL trained with the same iteration, or is trained still to be converged?

**Details Of Ethics Concerns:**

No Ethics Concerns.

---

> ### Author Response · Authors · 2023-11-19
>
> **Q1 More ablation studies are suggested to be conducted on more datasets.**
>
> **A1:** Thank you for your kind suggestions. The reason why we perform ablation studies on exposure correction can be explained in two folds: 1) The lightness among the different input images is quite different, which can be well-represented by our proposed CFTL mechanism. 2) The lightness adjustment directions of different exposures are quite different in this task. Therefore, this is a challenging task that can prominently evaluate how the proposed components contribute to performance improvement.
>
> As suggested, due to limited time, we provide more ablation studies on the low-light image enhancement tasks in the LOL dataset here. We have included the corresponding results in Table 5， and include the above discussion in the appendix, Sec. L.
> .
> | Configurations                    | PSNR/SSIM        |
> |----------------------------------|---------------------|
> | Baseline (DRBN)                     | 20.73/0.7986        |
> | +CFTL                        |   **23.71/0.8492**    |
> | +CFTL w/o global pooling               | 19.91/0.8317        |
> | +CFTL w/o channel-based ifft         | 22.66/0.8453        |
> | +CFTL w/o processing amplitude         | 22.78/0.8438        |
> | +CFTL w/o processing phase            | 23.29/0.8445        |
>
> **Q2 More visual comparisons should be provided.**
>
> **A2:** As suggested, we provide more results in the appendix, including visual results of low-light image enhancement (Fig. 24， Fig. 25, and Fix. 26), visual results of exposure correction (Fig. 27 and Fig.28), visual results of SDR2HDR on the HDRTV dataset (Fig. 29) and visual results of underwater image enhancement dataset on the UIEB dataset (Fig. 30).
>
> **Q3 Is the baseline and the baseline with the CFTL trained with the same iteration, or trained until it is converged?**
>
> **A3:** For the training configuration, we train the baseline and the baseline with the CFTL with the same iterations, and both of them converge for fair comparisons. We add more details in the Sec. A of the appendix.

---

### Official Review · Reviewer_WFwC · 2023-10-29

**Soundness:** 4 excellent
**Presentation:** 3 good
**Contribution:** 4 excellent
**Rating:** 6
**Confidence:** 5

**Summary:**

The authors introduces the channel-dimension Fourier transform for image enhancement. Its design is simple yet effective and comprise three straightforward steps.
The implementation formats is plug-and-play with previous image enhancement networks.
The authors experimentally evaluate the proposed model on various image enhancement tasks to depict its effectiveness.

**Strengths:**

(i) Overall, the idea is novel and interesting, and authors have clearly presented the motivation and algorithm with numerous figures and descriptions, making the principle of the algorithm easy to understand.
(ii) The core module design is simple and easy to implement. One particularly inspiring view of this algorithm is that it provides a new perspective of understanding global information representation.
(iii) The authors have also performed various experiments to validate the motivation, the effectiveness and efficacy of the algorithm, and the extensive application usage of the algorithm.

**Weaknesses:**

(i) Authors have performed experiments on various architectures, but some of them are a bit-of-date. In this way, I suspect if I just apply some other operations such as channel attention or spatial attention, the performance could also be improved a lot.
(ii) If I understand correctly, the improvement comes from the global information modeling. Nevertheless, most of baselines are CNN-based. Therefore, I doubt whether the algorithm would work when the baseline is set as the architecture with large receptive field.
(iii) Since authors provide various formats of the algorithm, and most of the formats achieve comparable results. I doubt if it is necessary to design so many formats for the usage.
(iv) The numerical results are improved with the algorithm, but the global information is easy to be affected. I doubt whether the improved version of the baseline has weak generalization ability than the baseline networks.

------------------------After Rebuttal---------------------------

Thank you for your feedback. I appreciate your acknowledgment of the partial alleviation of concerns. However, certain issues persist, giving the impression of overclaimed contributions in the paper. Here are specific points:

(i) The discussion on spatial and channel attention lacks comprehensiveness, omitting various attention mechanisms such as Fourier-based methods (e.g., GFNet). The application of standard spatial attention alone may not be sufficiently convincing.

(ii) The proposed method shows limited performance gains on baselines with a large receptive field. In the SNRformer results, only PSNR sees a slight improvement, casting doubt on the method's applicability in certain scenarios.

In conclusion, despite the novelty of the method, the rebuttal materials fall short of significantly demonstrating its compatibility with diverse frameworks and operations. Therefore, I have decided to lower the score.

**Questions:**

(i) I strongly suggest to include more contemporary method as the baseline methods.
(ii) I strongly suggest to discuss whether the generalization ability would be affected.
(iii) I suggest more tasks that related to image enhancement can be included for discuss, such as shadow removal, image harmony and style transfer.

---

> ### Author Response · Authors · 2023-11-19
>
> **Q1 Other channel attention or spatial attention can also elevate the performance.**
>
> **A1:** 1) We argue that some baselines are constructed based on spatial attention or channel attention, such as Restormer, which verifies our method can improve the performance on more contemporary architectures with this attention. 2) While we validate our method surpasses the global pooling like channel attention, the spatial attention would introduce more computation costs and is different from our proposed channel-dimension modeling. We present the results of adding spatial attention on the DRBN network in the LOL dataset here, which has a lower performance than our proposed method. We add the above discussion and results in Sec. K of appendix.
>
> | Settings| Baseline(DRBN) | +Spatial Attention | +Original CFTL (Ours) |
> |----|-----|-----|-----|
> | PSNR/SSIM   | 20.73/0.7986    | 21.05/0.8323  | 23.71/0.8492   |
>
> **Q2 Could the proposed method improve the performance of the architecture with the large receptive field?**
>
> **A2:** 1) We implement our method mainly on simple and lightweight architectures, which can facilitate the application of the simple architectures. Additionally, Restormer with a large receptive field can also be improved with our proposed method. 2) The mentioned architecture with a large receptive field enhances the global information with a huge computation burden, while our method is orthogonal to these methods that supplement a lightweight method. 3) Besides the Restormer, we also supplement another architecture SNRformer with the large receptive field as the baseline as illustrated in the response to Q5, the performance is also improved. We add the above discussion in Sec. K of the appendix.
>
>
> **Q3 Is it necessary to design various formats for implementation?**
>
> **A3:**  1) In terms of performance improvement, the variants of the CFTL can achieve performance improvement in most baselines. However, it is noticed that different formats of the CFTL achieve different performance gains in different baselines. Therefore, we cannot give a certain conclusion about which format can achieve the best performance when the baseline is unknown. The various formats provide alternatives. 2) In terms of extensibility, different formats provide different views of the channel-dimension Fourier transform. For example, the group CFTL illustrates a balance between the original CFTL and the global pooling with divided groups. Therefore, similar designs such as wavelet transform and fractional Fourier transform can also be included for implementation. While the high-order CFTL implies a potential to introduce more abstract information such as semantic information into the global vector. We add the above description in the Sec.I of the appendix.
>
> **Q4 Is the generalization ability would be degraded due to the global information is easy to be affected.**
>
> **A4:**  The generalization ability of image enhancement is not easily affected by global information. In fact, the mappings between different pairs are quite different in image enhancement tasks due to the inconsistency of the global information mapping (i.e., lightness), but they share a commonality in non-global information components (i.e., textures). Since the generalization ability is more dependent on the commonality component learning across different images, it would not be easily affected by global information. 2) We also provide the numerical results here, where we apply the model trained on LOL dataset to test on the Huawei dataset, and vice versa. The results suggest that the generalization performance remains constant with our proposed. We add the above discussion and the results in the Sec.H in the appendix.
>
> | Settings| LOL Train/ Huawei Test | Huawei Train/ LOL Test |
> |------|----|-----|
> | Restormer (Baseline)        | 19.50/0.6407    | 18.39/0.7697   |
> | +Pooling attention | 19.25/0.6454       |18.60/0.7415         |
> | +Spatial Fourier | 19.33/0.6436 |18.48/0.7141     |
> | +Original CFTL | 19.20/0.6430 | 18.62/0.7775|
> | +Group CFTL | 19.45/0.6444 | 18.98/0.7806 |
> | +High-order CFTL | 19.39/0.6460  | 18.19/0.7689  |
> | +Spatial-Fourier CFTL | 19.34/0.6453 | 18.36/0.7716  |

---

> ### Author Response · Authors · 2023-11-19
>
> **Q5 More newly proposed methods are suggested to be included as the baseline method.**
>
> **A5:** According to the suggestion, we supplement another method SNRformer as the baseline method due to limited time. We perform the experiments on the LOL dataset, and retrain the network under the same setting for fair comparisons. We supplement this result in Table 12 in the Sec. M in the appendix.
>
> | Settings | Baseline (SNRformer) | +Spatial Fourier | +Attention Pooling | +Original CFTL  | +Spatial-Fourier CFTL |
> |------------|---------|----------|-----------|--------|-------------|
> | PSNR/SSIM| 23.61/**0.8440** | 23.37/0.8375 | 22.47/0.8281 | **23.72**/0.8371 | 23.49/0.8386|
>
> **Q6 More tasks related to image enhancement are suggested to be included for discussion.**
>
> A6: 1) For the task such as shadow removal or image harmony, different channels reflect the different properties of the image. For instance, some channels are more related to the shadow regions, therefore, applying the channel-dimension Fourier-transform helps identify and enhance the representation for shadow information processing. 2) For the task such as style transfer, since the channel-dimension Fourier transform enhances the discriminability of the feature representation, the discriminability about the style can also be strengthened, thus it could provide an alternative space to conduct effective style transfer. We add this discussion in Sec.G in the appendix.

---

### Official Review · Reviewer_6DxW · 2023-10-30

**Soundness:** 4 excellent
**Presentation:** 3 good
**Contribution:** 3 good
**Rating:** 6
**Confidence:** 5

**Summary:**

1. A fresh perspective of Channel-dimension Fourier transform learning (CFTL) mechanism is proposed for image enhancement with three steps design.
2. Based on the CFTL mechanism , several usage formats are derived, which are compatible with existing methods.
3. The proposed approach showcases extensive ability across diverse image enhancement tasks with performance improvements over the baseline methods.

**Strengths:**

This paper brings a novel idea of channel-dimension Fourier transform with some strengths:
1. The overall framework is simple but fresh with introducing negligible computation costs.
2. The reasonableness of the approach’s design is presented with illustrations.
3. The experiments are sufficient to showcase the effectiveness of the introduced approach. Intermediate results are comprehensive to depict how the approach works.
4. The writing and organization are clear to follow.

**Weaknesses:**

However, there are several weakness/concerns need to be discussed, especially about some technique descriptions:
1. As far as I know, there is another work [1] attempts to introduce frequency design into channel dimension attention. Although the design and motivation behind the two works are different,  the relevance between the two works are not illustrated.
[1] FcaNet: Frequency Channel Attention Networks. ICCV 2021.
2. As a core claim, “The primary objective of CFTL is to capture global discriminative
representations by modeling channel-dimension discrepancy ”. What would this property help improve image enhancement are not well-explained.
3. As a core design space, the channel-dimension Fourier transform is applied in the global pooling-based space. The necessity of applying the core operation in this space are not fully discussed. What about using other spaces that possess global information properties?
4. As the concrete implementation, the Eq.(6) introduces the attention operation without many explanations.
5. Based on the above concerns, how to come up with the idea of the channel-dimension Fourier transform is not discussed, although its mechanism is illustrated.
6. Besides, more qualitative results are suggested to supplement in the appendix.

================================
1. The author's responses addressed my major concerns, however, the explanation of the "channel-dimension discrepancy" remains confusing. There is no explicit supporting evidence for this design. In fact, the implementations of "channel-dimension discrepancy" are diverse, such as feature orthogonalization [1][2], but these are not thoroughly discussed.

2. The diversity in sub-sequential filter responses caused by "channel-dimension discrepancy" seems to be more crucial for generalization performance [3]. The references provided by the author are related to all-in-one image restoration, which is a task more closely associated with generalization, but it fails to interpret the correlation between performance improvement and filter response diversity.

[1] Orthogonal Transformer: An Efficient Vision Transformer Backbone with Token Orthogonalization. NIPS 2022.

[2] OrthoNets: Orthogonal Channel Attention Networks. Arxiv 2023

[3] Reflash Dropout in Image Super-Resolution. CVPR 2022.

**Questions:**

Please see the above Weakness.

---

> ### Author Response · Authors · 2023-11-19
>
> Thank you for your comments and suggestions. We provide the point-to-point response below。
>
> **Q1 This work is similar to FcaNet, what is the relevance between these two works?**
>
> **A1:** The relationship between our work and FcaNet can be summarized in two folds. 1) Differences: FcaNet regards the global pooling in the channel attention as a special case of DCT transform. Then, it improves the global pooling by introducing higher-order DCT coefficients. Our method applies the operation for the global pooling vector with Fourier transform instead of expanding the global pooling, and could be integrated with FcaNet on its derived vector. 2) Similarity: both our work and FcaNet aim to construct the channel relationship from the frequency perspective. FcaNet derives the relationship by arranging different group channels with different frequencies, while our method unifies different channels in the frequency space to conjunct all channel information.
>
> **Q2 The reason why the “channel-dimension discrepancy” helps improve image enhancement is not illustrated.**
>
> **A2:** The reasons can be summarized in three aspects. 1) The powerful ability of neural networks is to convert the image information to high-dimension with discriminability. The channel-dimension Fourier transform can also enhance such ability to process information in high-dimension. 2) The channel-dimension relationship reflects the discriminative global information property that can contribute to image enhancement, which is similar to the conclusion in [1], where the polynomial integration of different channels enhances the image enhancement process. 3) The discriminability at the feature level also represents that the sub-sequential filter can respond to different features diversely, which has been proven in [2], leading to the improvement of image processing. We add these descriptions in Sec. G of the appendix.
>
> **Q3 Why implement the channel-dimension Fourier transform in the global pooling space? What about using other spaces that possessed global information property?**
>
> **A3:** The reasons of performing the implementation in the global pooling space can be summarized in two aspects. 1) As shown in Fig.2, we can conclude the global pooling space comprises the main information about the global lightness and color information, and other information about the content can be excluded. Thus, performing the implementation in the global pooling space makes the operation focus on the global component adjustment. 2) The global pooling space also provides a convenient space for conducting operations, where information can be easily affected in this space with only 1x1 convolution in single-dimension space, while the original space can hardly be affected with simple convolution operations.
>
> For other spaces with global information property, we have verified the effectiveness of applying the channel-dimension Fourier transform in the Fourier space (spatial-Fourier CFTL), which possesses global information according to spectral theory [3]. Moreover, for other spaces, since the Gram matrix derived from the self-attention is already calculated along the channel-dimension, applying our method may not work well.
>
> **Q4 Why introduce the attention operation in the Eq.(6) is not explained.**
>
> **A4:** The reasons can be attributed in two aspects: 1) This operation simulates the channel-attention operation, leading to effective processing of the global information.
> 2) This operation also caters to the global matrix transformation such as color transformation, satisfying the requirement of image enhancement.

---

> ### Author Response · Authors · 2023-11-19
>
> **Q5 What is the motivation behind this work is not illustrated, why propose this method?**
>
> **A5:** The motivation behind this work can be attributed to three aspects. 1) Channel dimension. The channel-dimension relationship reflects the feature information property. For some tasks such as the style transfer, the Gram matrix can reflect the style information effectively. Therefore, we propose the channel-dimension Fourier transform that provides an alternative format to construct the relationship of channel dimension, which reflects the property like global style information. 2) Fourier transform. Previous methods (i.e., FFC[1], GFNet[2] ) have verified the effectiveness of conducting the operation in the Fourier space, which processes the global information conveniently referring to the spectral theory [1]. Meanwhile, applying the operation in the channel-dimension Fourier space has not been fully explored, which can also affect the channel-dimension information effectively and conveniently. 3) Global information processing. Since the global information is strongly related to image enhancement as illustrated in Fig 2 in the main paper, the channel-dimension Fourier can enhance it to the high-dimension space, leading to an effective process of the above global information and thus improving image enhancement performance. We add the above discussion and more analysis in Sec.D of the appendix.
>
> **Q6 More visual results are suggested to provide.**
>
> **A6:** Thanks for your advice. In the appendix, we present additional findings, including visual results of low-light image enhancement (Fig. 24， Fig. 25, and Fix. 26), visual results of exposure correction (Fig. 27 and Fig.28), visual results of SDR2HDR on the HDRTV dataset (Fig. 29) and visual results of underwater image enhancement dataset on the UIEB dataset (Fig. 30).
>
>
> [1] Mustafa A, Hanji P, Mantiuk R. Distilling style from image pairs for global forward and inverse tone mapping. CVMP 2022.
>
> [2] Park D, Lee B H, Chun S Y. All-in-one image restoration for unknown degradations using adaptive discriminative filters for specific degradations. CVPR 2023.
>
> [3] Chi L, Jiang B, Mu Y. Fast Fourier convolution. In NeurIPS 2020.
>
> [4] Rao Y, Zhao W, Zhu Z, et al. Global filter networks for image classification. In NeurIPS 2021.

---

### Meta-Review · Program_Chairs · 2023-12-05

**Metareview:**

This meta-review is written by the Program Chairs.

After calibration and downweighting non-informative and inflated reviews, the assessment is that this paper does not meet the accept threshold.

The paper's contributions are incremental.  A notable issue is that the authors did not discuss connections to this very closely related paper: https://proceedings.mlr.press/v202/zhou23f/zhou23f.pdf
(The authors should have been well aware of this paper.)

**Justification For Why Not Higher Score:**

Please refer to the metareview.

**Justification For Why Not Lower Score:**

Please refer to the metareview.

---

### Decision · Program_Chairs · 2024-01-16

Reject